# Autumn destabilization of deep porewater CO$_2$ store in a northern peatland driven by turbulent diffusion

A. Campeau [1,2✉], D. Vachon[3], K. Bishop [4], M. B. Nilsson [1] & M. B. Wallin [2,4]

The deep porewater of northern peatlands stores large amounts of carbon dioxide (CO$_2$). This store is viewed as a stable feature in the peatland CO$_2$ cycle. Here, we report large and rapid fluctuations in deep porewater CO$_2$ concentration recurring every autumn over four consecutive years in a boreal peatland. Estimates of the vertical diffusion of heat indicate that CO$_2$ diffusion occurs at the turbulent rather than molecular rate. The weakening of porewater thermal stratification in autumn likely increases turbulent diffusion, thus fostering a rapid diffusion of deeper porewater CO$_2$ towards the surface where net losses occur. This phenomenon periodically decreases the peat porewater CO$_2$ store by between 29 and 90 g C m$^{-2}$ throughout autumn, which is comparable to the peatland's annual C-sink. Our results establish the need to consider the role of turbulent diffusion in regularly destabilizing the CO$_2$ store in peat porewater.

[1] Department of Forest Ecology and Management, Swedish University of Agricultural Sciences, Umeå, Sweden. [2] Department of Air, Water and Landscape, Uppsala University, Uppsala, Sweden. [3] Department of Ecology and Environmental Sciences, Umeå University, Umeå, Sweden. [4] Department of Aquatic Sciences and Assessment, Swedish University of Agricultural Sciences, Uppsala, Sweden. ✉email: audrey.campeau@slu.se

Northern peatlands represent an important long-term global sink of atmospheric carbon dioxide ($CO_2$) that has contributed to cool the Earth's atmosphere throughout the Holocene[1–3]. This C sink arises from the primary production at the peatland surface, being higher than C mineralisation cumulated over the entire peat depth, which leads to C burial in the form of peat. Peatlands are typically viewed as two-layer systems[4]. They feature a highly active layer (acrotelm), which comprises the peatland's surface (i.e. above the lowest water table), and an underlying layer (catotelm) of permanently waterlogged peat[5]. Most of the peatland $CO_2$ cycling is considered to operate within the acrotelm[6]. There, living plants fix $CO_2$ from the atmosphere and oxygen is freely available to fuel respiration that returns a fraction of this $CO_2$ to the atmosphere. Water moves relatively rapidly through the partially decomposed peat in the acrotelm, hence, also generating most of the lateral $CO_2$ export in runoff[7–10]. In comparison, the catotelm peat layer is generally cold and void of oxygen, which slows down decomposition[6]. The catotelm porewater is also constrained by, in general, low hydraulic conductivity with deeper, more decomposed and compacted peat[11]. Water in the catotelm porewater is typically rich in dissolved $CO_2$, as a consequence of slow but relatively constant $CO_2$ production coupled with even slower removal processes[12–14]. As long as $CO_2$ remains confined in the catotelm, its role in the peatland $CO_2$ cycling is negligible.

Transport of porewater $CO_2$ from the catotelm to the acrotelm is mostly driven by diffusion[12,15,16]. This vertical diffusive transport is presumed to take place through molecular diffusion, a slow and constant process (diffusion coefficient $D \sim 10^{-9}$ to $10^{-8}$ $m^2 s^{-1}$, depending on peat porosity and tortuosity[12,17]). Under strict vertical molecular diffusion, a molecule of $CO_2$ generated a meter below the water table will take about 15 years to reach the acrotelm, where net losses by atmospheric $CO_2$ emission or lateral hydrological $CO_2$ export can take place. Hence, there is a general agreement that the catotelm peat porewater $CO_2$ store contributes little to the seasonality in peatland $CO_2$ cycling[18]. Studies documenting catotelm porewater $CO_2$ dynamics in northern peatlands have hitherto relied solely on methodologies with a low temporal sampling resolution (e.g. refs. [19–21]). The assumption of slow dynamics in catotelm porewater gas store has not been explicitly tested, which may result in overlooking key process controlling temporal dynamics in peatland $CO_2$ cycling.

Here, we evaluate the stability of the catotelm porewater $CO_2$ store in a boreal peatland using automated in situ sensor technology. Our data consist of hourly measurements over 4 consecutive years of porewater $CO_2$ concentration and temperature at different depths in a 2-m-deep vertical peat profile. These data reveal rapid and regular losses of catotelm porewater $CO_2$ recurring every autumn. We assess the rate of vertical diffusion, based on the heat budget method, and demonstrate that diffusion in the catotelm porewater occurs at orders of magnitude above the molecular rate of diffusion. Vertical diffusive transport occurs instead through turbulent diffusion, which reflects the presence of small-scale random fluid motion propagating through the peat porewater[22]. The regular weakening of porewater thermal stratification every autumn enhances turbulent diffusion, which results in rapid transport of deep porewater $CO_2$ towards the surface. These findings reveal a hitherto unknown process that makes the catotelm and its porewater $CO_2$ store far more dynamic than previously thought. The implications for the peatland $CO_2$ cycle, however, remain to be fully investigated.

## Results and discussion
**Porewater $CO_2$ and temperature dynamics.** Porewater $CO_2$ concentration timeseries reflect the constantly evolving balance between input and removal processes. The $CO_2$ inputs in peat porewater occur through transport from adjacent layers and from local biogenic processes (e.g. rhizospheric and microbial production). The latter is mostly controlled by temperature, organic matter quality and oxygen or other electron acceptors availability[23]. Losses of porewater $CO_2$ occurs mostly by transport processes that remove $CO_2$ from the porewater (e.g. atmospheric emission and lateral hydrological export). Here we found that porewater $CO_2$ concentration was lowest and most variable in the shallowest porewater (0.13 m in Fig. 1a, Fig. 2a), which is consistent with dynamic equilibrium between inputs and removal processes. Porewater $CO_2$ concentration increased and became steadier with depth (Figs. 1a, 2), which is again consistent with slower and more constant input and removal processes. There were, however, events of large and sudden losses in porewater $CO_2$ concentration recurring every autumn over the 4 consecutive years of observation that suggested a sudden rise in transport processes (Fig. 1a).

These events consisted of abrupt sequential decreases in peat porewater $CO_2$ concentration along the peat depth profile. The large decreases in $CO_2$ concentration first occurred in the surface porewater (0–0.25 m) between the end of August to September, and progressed down to deeper porewaters (0.75 m), around early October, and occasionally at 1.5 m depth later in November (Fig. 1a). The decreases in porewater $CO_2$ concentration were most dramatic and regular at 0.75 m depth, only occurring near the end of September to mid-October (Fig. 1a). Outside of autumn, an isolated event of rapid decrease in $CO_2$ concentration in the shallowest porewater (0.13–0.38 m depth) was identified between June 5th and 11th 2016. In early winter, the peat porewater $CO_2$ concentration (only measured at 0.75 and 1.5 m depth) steadily returned to average growing season levels (Fig. 1a, Supplementary Fig. 1 and Supplementary Table 1) and remained relatively stable throughout winter and spring.

Periods of rapid decrease in porewater $CO_2$ concentration always occurred together with the equilibration of porewater temperatures between two adjacent depths (Fig. 1a, b). The weakening of thermal stratification between water masses is associated with fluid instability. This instability can be expressed by the buoyancy frequency ($N^2$, $s^2$)[24], whereby weak thermal stability between two porewater layers is indicated by $N^2$ approaching zero (Fig. 1c). Periods where the $N^2$ was near-zero coincided with periods with the most variable porewater $CO_2$ concentrations (Fig. 2), hence reinforcing a link between the weakening of porewater thermal stability and losses from the $CO_2$ store. Each event of rapid loss in porewater $CO_2$, including those in autumn and in June 2016, corresponded with periods where $N^2$ was near-zero (Fig. 2). The event in June 2016 occurred due to unusually cold weather conditions that equilibrated the surficial peat porewater temperatures for about 6 days (Fig. 1a–c).

Throughout much of the ice-free season, porewater temperatures were considerably warmer near the surface and decreased sharply with depth, thus generating a strong vertical thermal stratification (high thermal stability, high $N^2$) (Fig. 1b, c). In autumn, the progressive equilibration of porewater temperature along the peat depth profile occurs due to surface cooling, which brings $N^2$ near zero and weakens the thermal stability (Fig. 1b, c). The vertical equilibration of catotelm porewater temperature begins in the surficial porewater in mid-August [0.13–0.25 m] and progresses down the depth profile in October and November [0.75–1.5 m] (Fig. 1c), following the same sequence as the sudden losses of porewater $CO_2$ concentration (Fig. 1a).

The large and rapid decreases in porewater $CO_2$ concentration during periods of weak thermal stability dramatically altered the shape of the porewater $CO_2$ depth profile. Throughout most of the growing season (high thermal stability, high $N^2$), the

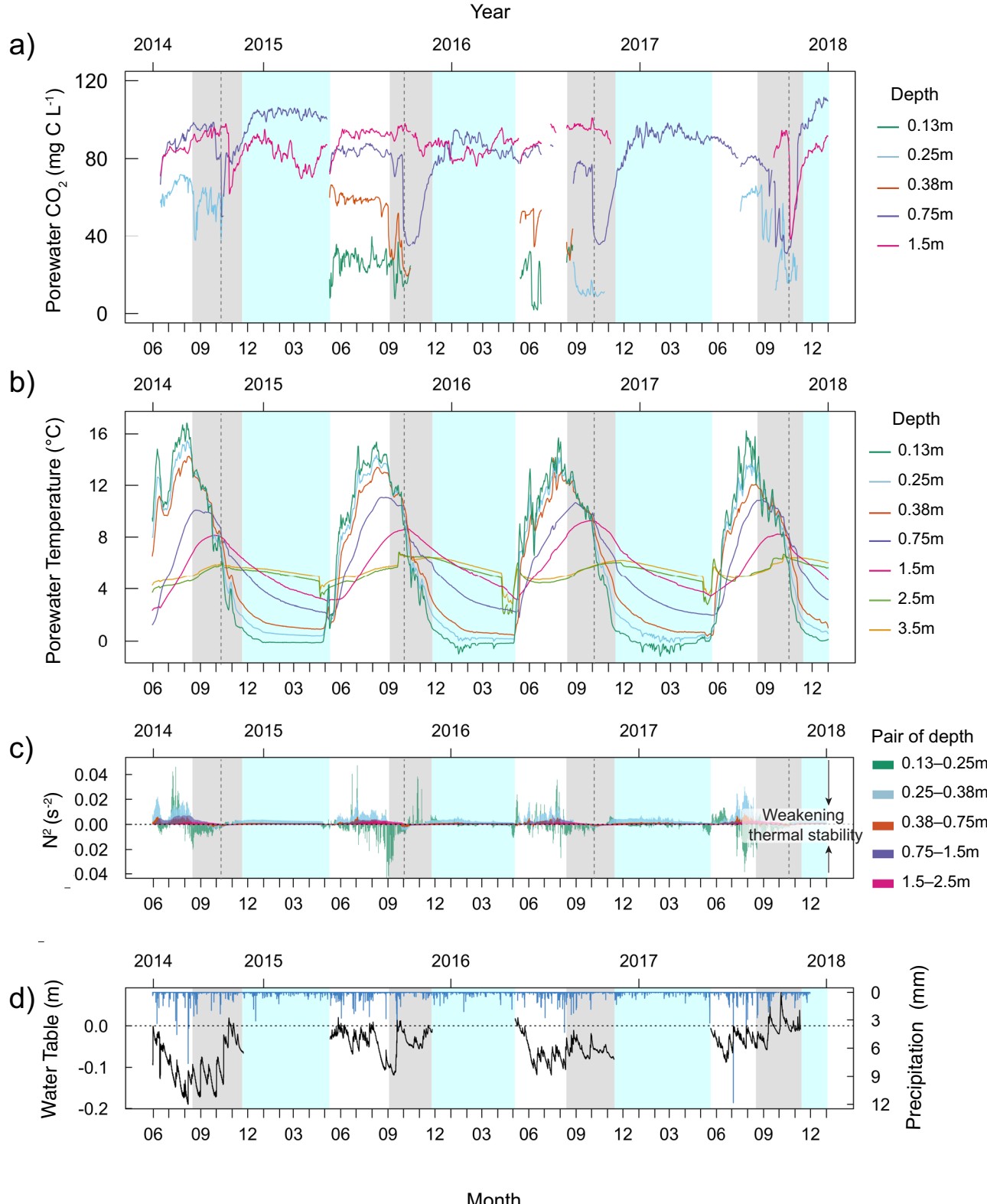

porewater $CO_2$ concentration increased sharply with depth, which resulted in a persistent convex profile from May to August (Fig. 3a). This convex vertical $CO_2$ concentration gradient is consistent with slow vertical diffusive transport[17]. During the autumn, the porewater $CO_2$ depth gradient suddenly became linear with depth (Fig. 3b) or sometimes collapsed completely between two adjacent porewater depths (Fig. 3d). There were also

occasional inversions in the porewater $CO_2$ depth gradient in autumn, during which $CO_2$ concentration became higher in shallower porewater than in the deeper porewater (Fig. 3c). For example, the sudden decreases in porewater $CO_2$ concentration at 0.75 m sometimes coincided with a brief increase in the surficial porewater $CO_2$ (depths from 0.38 to 0.13 m) (Fig. 1a). The decrease in the $CO_2$ depth gradient (going from convex to linear),

**Fig. 1 Porewater $CO_2$ concentration, temperature, buoyancy frequency and hydrological conditions over time.** Timeseries of the porewater **a** $CO_2$ concentration (mg C $L^{-1}$) and **b** temperature (°C) at each depth, **c** buoyancy frequency ($N^2$, $s^{-2}$) between pairs of adjacent porewater depths and **d** water table position (metres relative to ground surface) (black line) and hourly precipitation (mm) (blue lines), from June 2014 to January 2018. In **a** and **b**, each line represents a different depth relative to ground surface (0.13 m (green), 0.25 m (blue), 0.38 m (orange), 0.75 m (purple) and 1.5 m (magenta) below ground surface). In **b**, porewater temperature measurements at 2.5 m (light green) and 3.5 m (yellow) are also presented. In **c**, each coloured area represents a different pair of porewater depths 0.13 and 0.25 m (green area), 0.25–0.38 m (blue area), 0.38–0.75 m (orange area), 0.38–0.75 m (orange area), 0.75–1.5 m (purple area) and 0.75–1.5 m (purple area), 1.5–2.5 m (magenta area). The vertical dotted lines in (**a–c**), marks the day of weakest porewater thermal stability (i.e. equal temperatures from 0 to 1.5 m deep). In each panel, the x-axes indicate the dates, with years on top and month numbers on the bottom axis. The background areas, coloured in grey and cyan, mark the periods of weak thermal stability and ice/snow cover on the peatland surface, respectively.

together with the sequential fluctuations over the depth profile and occasional inversions strongly suggest a sudden increase in vertical diffusive transport of catotelm porewater $CO_2$ towards the surface during periods of weakened thermal stability.

**A new consideration of turbulent diffusion.** The direction of diffusive transport is dictated by concentration gradients (i.e. diffusing from areas of highest to lowest concentration). The speed at which this diffusion occurs, whether molecular or turbulent, depends on myriad environmental conditions. The molecular diffusion occurs at a low and constant speed (coefficient below $10^{-7}$ $m^2$ $s^{-1}$ [12,15,17]), while the turbulent diffusion is faster and can varyorders of magnitude in both time and space (coefficient above $10^{-7}$ $m^2$ $s^{-1}$)[25]. Vertical diffusive transport of $CO_2$ in the catotelm porewater is considered to take place mostly by molecular diffusion[12,15,16]. However, our estimates of the apparent diffusion coefficient ($K_{app}$) in the peat porewater, based on the heat budget method[26], occurs at orders of magnitude higher than is possible by molecular diffusion and varies widely over time (i.e. from $10^{-7}$ to $10^{-4}$ $m^2$ $s^{-1}$ (Fig. 4a)). Thus, we surmise that turbulent diffusion governs the vertical diffusive transport of catotelm porewater $CO_2$. This implies that vertical $CO_2$ diffusion is enhanced relative to the molecular rate by the presence of small-scale random fluid motion (i.e. turbulence) in the catotelm porewater.

According to Osborne[24], the turbulent diffusion coefficient ($K_z$) can be described as:

$$K_z = \gamma_{mix}\varepsilon/N^2 \qquad (1)$$

where the turbulent diffusion coefficient ($K_z$; $m^2$ $s^{-1}$) varies as a function of the kinetic energy dissipation rate ($\varepsilon$; $m^2$ $s^{-3}$) divided by the strength of the water density stratification (buoyancy ($N^2$), $s^{-2}$). In freshwater, buoyancy mostly reflects changes in vertical water temperature stratification. The mixing efficiency ($\gamma_{mix}$) describes the fraction of energy storage as potential energy[24,27]. The kinetic energy production (here as equivalent to dissipation at steady state) provides a source of turbulence, but the strengthening and weakening of porewater thermal stability ($N^2$) determine the degree of suppression or propagation of this turbulence (Eq. (1)[24])[22]. We observed an increase in the $K_{app}$ with decreasing $N^2$, a relationship that is consistent with the model proposed by Osborne[24] and previous observations from lakes[27,28] (Eq. (1), Fig. 4b). The kinetic energy in the catotelm porewater was overall low ($10^{-7}$ to $10^{-9}$ $m^2$ $s^{-3}$; Fig. 4b), which is about one order of magnitude lower than in small and sheltered northern lakes[29,30]. The kinetic energy was nonetheless higher in the near-surface porewater and attenuated with depth (Fig. 4b). This kinetic energy is likely supplied from wind shear near the surface and lateral water flow through the peat pores.

Even under constant and low kinetic energy inputs, there was a three order of magnitude shift in the $K_{app}$ with changing porewater thermal stability over time ($N^2$), (Fig. 4b, Eq. (1)). The $K_{app}$ ranged from $10^{-7}$ to $10^{-6}$ $m^2$ $s^{-1}$ during periods of strong thermal stability

(May to August, high $N^2$), to $10^{-5}$ to $10^{-4}$ $m^2$ $s^{-1}$ during periods of weak thermal stability (August to November, $N^2$ near zero) (Figs. 4 and 5). The seasonality in turbulent diffusion implies that a molecule of $CO_2$ generated 1 m below the water table can diffuse to the surface in just 1–2 h during periods of weak thermal stability (autumn), compared with 1.5 years during periods of strong thermal stability (growing season), and 15 years solely via molecular diffusion (Fig. 5). Combining the increase in turbulent diffusion in autumn with the steep vertical porewater $CO_2$ concentration gradient that builds up over the growing season may lead to a rapid diffusion of catotelm porewater $CO_2$ towards the surface where net losses occur.

Vertical equilibration of porewater temperature across the almost entire depth profile is unique to the autumn season and leads to a particularly weak porewater thermal stability (Fig. 1b). During most of the growing season, the surficial porewater is considerably warmer than the lower depths resulting in strong thermal stratification and stability. During winter and spring, an inverse vertical porewater temperature stratification is maintained by colder porewater near the ice and snow cover at the surface (Fig. 1b). Snow and ice cover throughout winter also shelters the porewater from kinetic energy input induced by wind or lateral flow, which contributes to suppressing turbulence in the peat porewater and increase stability in the porewater $CO_2$ store. Hence, we suggest that unique conditions of weak porewater thermal stability in autumn lead to a sharp increase in turbulent diffusion that causes the recurring destabilizations and losses of the catotelm porewater $CO_2$ store each autumn. A rapid diffusive transport of catotelm porewater $CO_2$ towards the surface could enhance atmospheric $CO_2$ emission and lateral hydrological $CO_2$ export, causing the observed periodic loss of catotelm porewater $CO_2$ store each autumn.

The recovery of porewater $CO_2$ concentration in early winter was generally slow and steady (average rate of increase 0.05 (SD ± 0.12) and 0.04 (SD ± 0.07) mg C $g^{-1}$ $d^{-1}$ at 0.75 and 1.5 m, respectively, across all 4 years combined (Supplementary Table 1)). The return of porewater thermal stability and the onset of soil frost in winter decreases the vertical diffusion of porewater $CO_2$ in early winter, which can lead to a recovery in concentration through local biogenicl production. The observed rate of recovery in porewater $CO_2$ concentration during early winter is consistent with measurements of biogenic $CO_2$ production in laboratory incubations at cold temperatures[31–33]. Similarly steady recoveries have been reported in northern lakes following ice-cover formation[34–36]. There were, however, days in 2014 and 2017 where the increase in the porewater $CO_2$ concentration at 0.75 m was more sudden (rate between 0.2 and 1.2 mg C $g^{-1}$ $d^{-1}$) (Supplementary Fig. 1). The autumn of 2014 and 2017 were the only years where large and sudden losses in porewater $CO_2$ occurred at 1.5 m, indicating that the destabilization of porewater $CO_2$ store reached deeper depths in those two years. We therefore consider that those isolated days of more rapid recovery in porewater $CO_2$ concentration at 0.75 m resulted

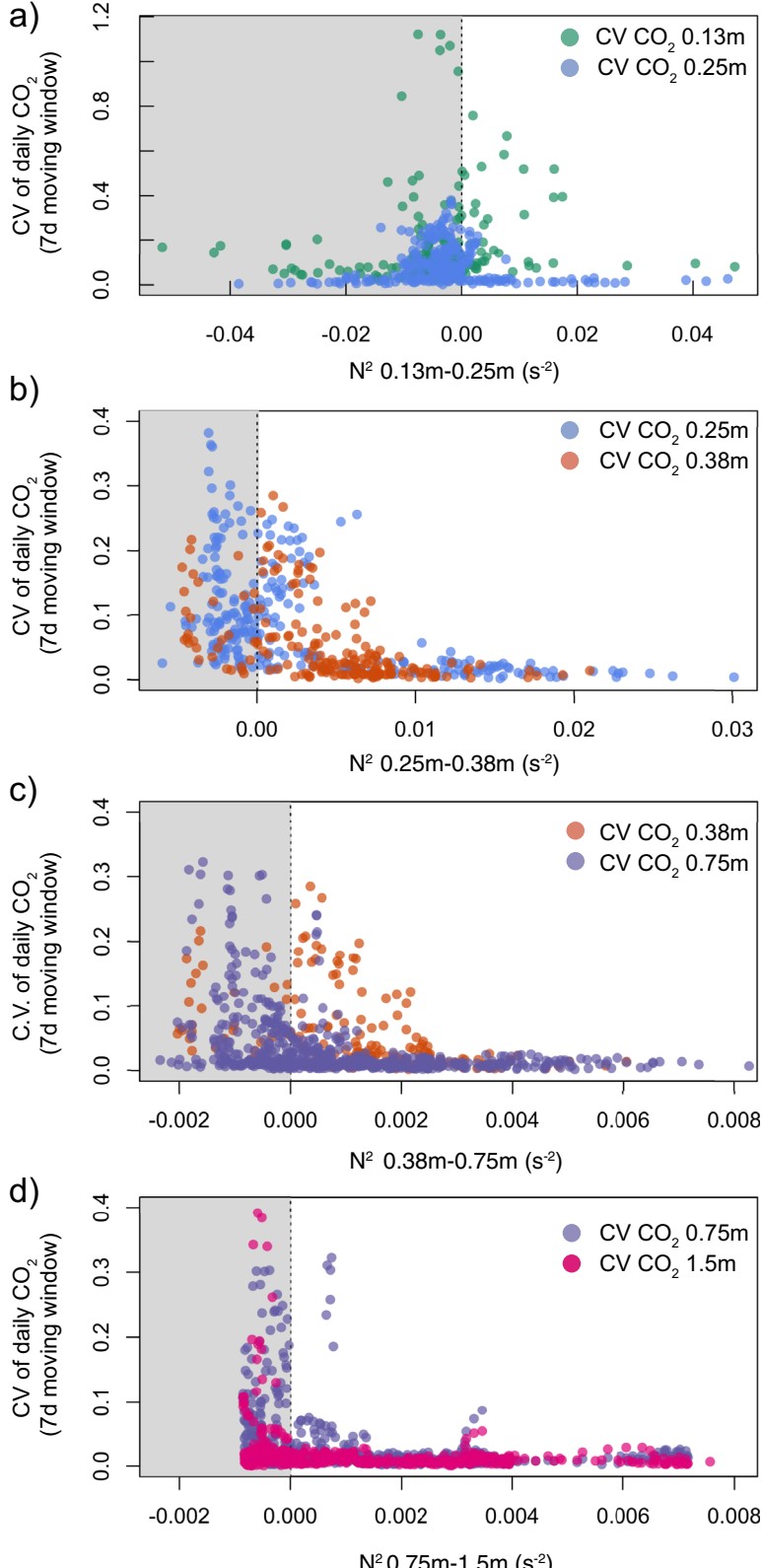

**Fig. 2 Relationship between porewater CO$_2$ concentration variability and thermal stability.** The 7-day moving coefficient of variation (C.V) for the daily averaged porewater CO$_2$ concentration plotted against the daily averaged buoyancy frequency (N$^2$) between different pairs of adjacent porewater depths: in **a** 0.13 and 0.25 m, **b** 0.25 and 0.38 m, **c** 0.38 and 0.75 m, **d** 0.75 and 1.5 m.

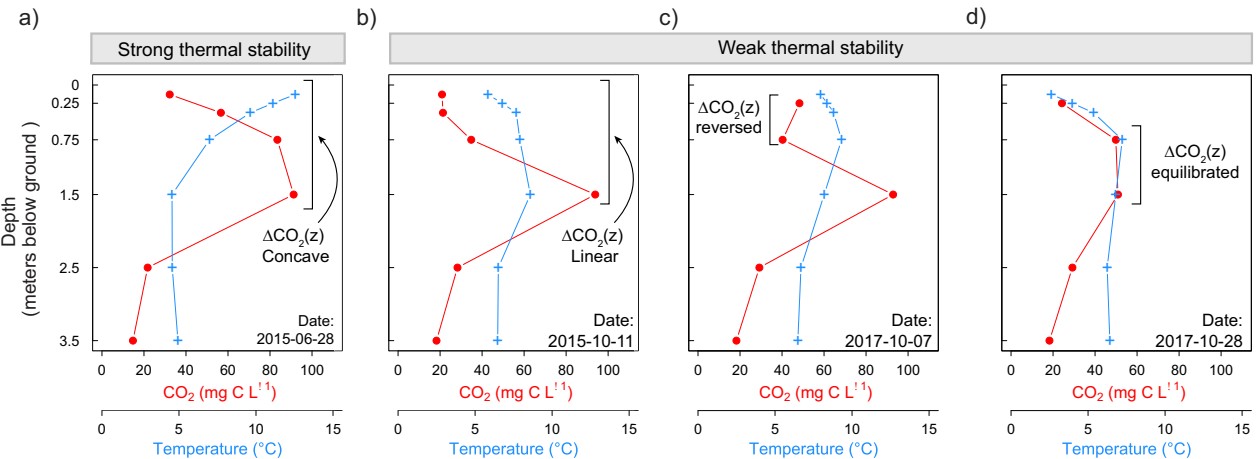

**Fig. 3 Shapes of the peat porewater $CO_2$ concentration depth profile.** Depth profile of the daily averaged porewater $CO_2$ concentration (red circles and lines) and temperatures (blue crosses and lines) from 0 to 3.5 m below ground surface. Each panel presents a different example of depth profiles: **a** during strong thermal stability, **b** weak thermal stability, where the $CO_2$ depth gradient ($\Delta CO_2(z)$) is linear, **c** weak thermal stability, where the $\Delta CO_2(z)$ between 0.25 and 0.75 m is reversed, and **d** weak thermal stability, where the $\Delta CO_2(z)$ between 0.75 and 1.5 m is in equilibrium (i.e. almost null).

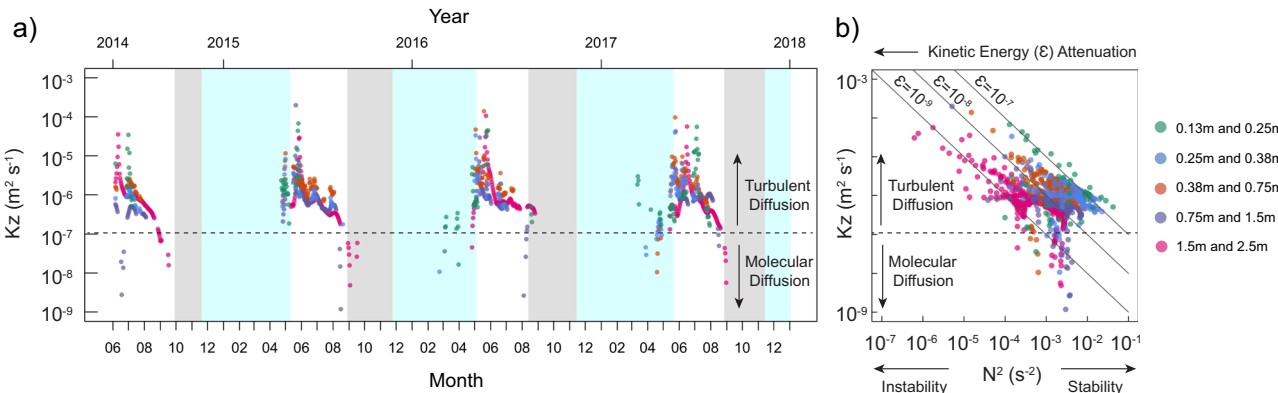

**Fig. 4 Vertical diffusion coefficient over time and as a function of thermal stability. a** Timeseries of the apparent diffusion coefficient ($K_{app}$; m$^2$ s$^{-1}$) between two adjacent porewater depths between June 2014 and January 2018, derived from the heat budget method and **b** scatterplot of the $K_{app}$ in relation to the buoyancy frequency (N$^2$; s$^{-2}$) between the same pairs of porewater depths. Symbols are coloured based on the pairs of porewater depths, 0.13 to 0.25 m (green), 0.25 to 0.38 m (blue), 0.38 to 0.75 m (orange), 0.75 to 1.5 m (purple), 1.5 to 2.5 m (magenta). The horizontal dotted lines in both **a** and **b** mark the threshold between the turbulent and molecular rate of diffusion. Oblique full lines in **b** indicate the theoretical relationship between $K_z$ and N$^2$ described by Eq. (1)[24], under different levels of kinetic energy input ($\varepsilon = 10^{-7}$ to $10^{-9}$ m$^2$ s$^{-3}$) and a constant $\gamma_{mix}$ of 0.10. In **a**, the x-axes indicate the dates, with years on top and month numbers on the bottom axis. In **a**, the background areas coloured in grey and cyan mark the periods of weak thermal stability and ice/snow cover on the peatland surface, respectively.

from additional transport processes from subjacent or adjacent porewater and were not strictly explained by local biogenic production.

Turbulent diffusion has been studied extensively in lakes, where diffusion at the molecular level is almost always exceeded by turbulence in free-flowing water[27,28,37]. Turbulence is almost certainly on smaller scales in the catotelm porewater than in open water bodies (probably much lower than mm compared with mm to cm-scale eddies in lakes) (Fig. 5b, c). The increase in turbulent diffusion proposed in this study as a mechanism for the destabilization of porewater $CO_2$ store in autumn should not be confused with the convective mixing that occurs in northern lakes in autumn (i.e. autumn lake turnover). While substantial losses of the dissolved $CO_2$ store may occur in both lakes and peatlands during autumn, the hydrology of peatlands is critically different to that of a lake. In the case of lakes, the weakening of thermal stability in autumn also results in an increase in turbulent diffusion, but this increase is overridden by an increase in convection and advection[22]. The increased mass flow overturns

the whole water column, causing large amounts of $CO_2$ to be released to the atmosphere[28,36]. In peat porewater, mass flow is constrained by the low hydraulic conductivity and strong anisotropy of the peat. Hence, thermally driven convective mixing in peat porewater is only possible in areas where hydraulic conductivity is orders of magnitude higher than those measured across our studied peatland[38] (Supplementary Fig. 3). Monthly measurements of the stable isotope ratio of porewater ($\delta^{18}O$) at this site confirms a persistent stratification of water masses across this depth profile in autumn (Supplementary Fig. 6). This lasting stratification does not indicate a simultaneous convective mixing with the increase in turbulent diffusion in autumn.

Some properties of the studied location could have made this peat depth profile prone to higher levels of turbulence in the catotelm porewater. The bulk density of the peat is slightly lower (0.016 g cm$^{-3}$, Supplementary Fig. 2) than in most northern peatlands (range 0.02–0.25 g cm$^{-3}$ [39]) and other locations within the studied peatland (0.05 g cm$^{-3}$ [40]). The low bulk density of the

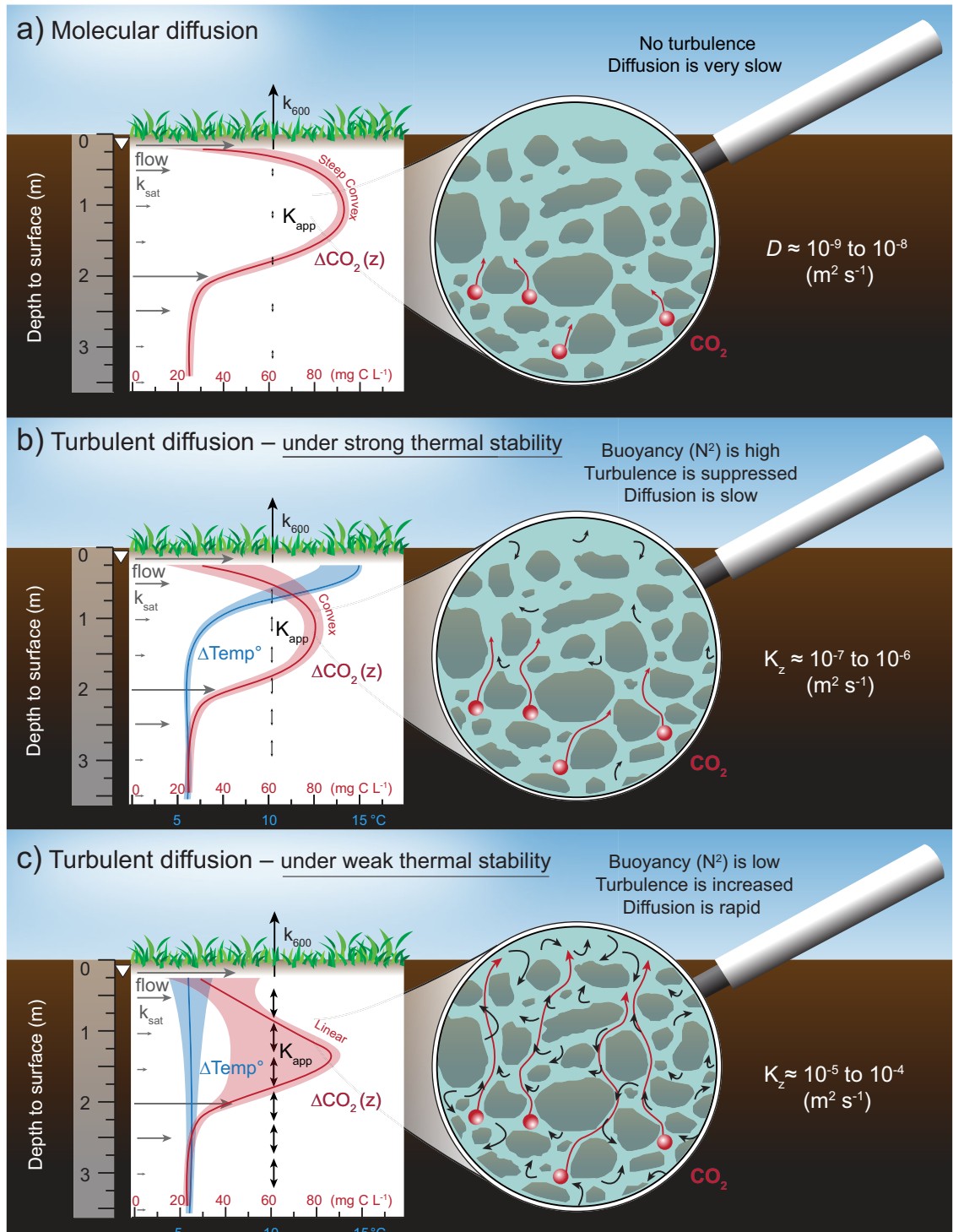

**Fig. 5 Schematic of the different rates of vertical $CO_2$ diffusion in the catotelm porewater.** From slowest to fastest: in **a** molecular diffusion, **b** turbulent diffusion during periods of strong porewater thermal stability (e.g. growing season) and **c** turbulent diffusion during periods of weak porewater thermal stability (e.g. autumn). Each panel shows the depth profile of porewater $CO_2$ concentration (red line and area) and temperature (blue line and area), with the presumed saturated hydraulic conductivity ($k_{sat}$, grey arrows) based on bulk density (Supplementary Fig. 2) and Campeau et al.[13]. Each panel also includes a magnifying glass illustrating the scale of vertical porewater $CO_2$ diffusion (red spheres and arrows) and turbulence (black arrows) in the catotelm porewater. In **a**, the apparent diffusion coefficient ($K_{app}$) is at the molecular rate, which is slow and constant over time (below $10^{-7}$ $m^2$ $s^{-1}$), resulting in a steep convex porewater $CO_2$ depth gradient ($\Delta CO_2(z)$). In **b**, turbulence is present in the porewater, but it is suppressed by the strong thermal stability (high buoyancy ($N^2$)), which results in a slow vertical $CO_2$ diffusion, nonetheless exceeding the molecular rate, yielding a persistently convex $\Delta CO_2(z)$. In **c**, the weak thermal stability (low $N^2$, $K_{app}$ increases) allows turbulence to propagate in the catotelm, which enhances vertical porewater $CO_2$ diffusion (upwards or downwards), potentially leading to variable $\Delta CO_2(z)$.

peat is associated with a higher porosity (98%), which could potentially increase the propagation of turbulence in the catotelm. Our studied location is also found in a flow convergence zone and in proximity with the stream initiation point at the mire outlet, which may lead to high lateral water flow through the peat profile, supplying an additional source of kinetic energy production. Furthermore, a preferential flow path has been previously documented between 2 and 2.5 m depth[13], suggesting potential further sources of kinetic energy production deep in the catotelm. However, these deep preferential flow paths, pipes and/or macropores are relatively common features in peatlands[41–43]. Thirdly, our studied peatland forms a large mire complex that is located at a topographic high point within the landscape, which might contribute to strong wind exposure (mean 2.6 m s$^{-1}$, max 12.3 m s$^{-1}$ over the full study period) that supplies additional kinetic energy and turbulence to the catotelm porewater.

**Other drivers of porewater $CO_2$ dynamics**. Other factors, such as changes in biological $CO_2$ production, water table position, air-water gas exchange velocity and ebullition contribute to the temporal variability in porewater $CO_2$ concentration. However, we consider those factors to have a relatively minor effect on the regular and rapid losses in catotelm porewater $CO_2$ store each autumn compared with the increase in turbulent diffusion associated with weakening porewater thermal stability. Given the inhospitable conditions in the catotelm (e.g. energy substrate limitation and low amplitude in annual temperature), the apparent temperature response of peat decomposition is generally linear[44–46] and sometimes inconsistent due to shifting metabolic pathways[33]. Annual temperature in the deep porewater vary across a narrow range (1–11 °C at 0.75 m and 2–9 °C at 1.5 m) with annual maxima being reached between August and October (Fig. 1b). It appears unlikely that the small and gradual changes in deep porewater temperature in autumn can cause the sudden losses in porewater $CO_2$ observed in our data. Ecosystem respiration measurements at this site also indicate that substantial $CO_2$ production still takes place across autumn despite the cooling temperatures (Fig. 6). Furthermore, the observed rate of loss in deep porewater $CO_2$ store in autumn exceeds even the highest measurement of peat decomposition[23], indicating that a complete shutdown of $CO_2$ production cannot possibly explain the observed losses in catotelm porewater $CO_2$ concentration. Together, this indicates that a possible decline in porewater $CO_2$ production in autumn cannot fully explain the phenomenon present in our data.

Changes in water table position, which can be associated with $CO_2$ dilution by rain water or changes in porewater flow direction, appear negligible in explaining the timing, depth, and magnitude of the changes in porewater $CO_2$ store each autumn (Fig. 1d). Changes in water table position showed a rising, declining or stable water table at the time of the rapid drop in porewater $CO_2$ concentration (Fig. 1d). The largest water table change represented a maximum 10% recharge of the total porewater volume contained in the top 2 m of the peat profile. In comparison, a refilling of 23 to 100% of the top 2 m of the peat porewater by $CO_2$-poor rainwater would be needed to generate the observed drop in porewater $CO_2$ store in autumn by dilution. While several studies have presented evidence of deep porewater recharge, with considerable influence on solutes concentration peat depth profiles[47–49], the monthly measurements of porewater $\delta^{18}O$ value at this site indicated only shallow rainwater infiltration (e.g., summer storm in August 2015 ($^{18}O$ enriched rain at 0.13 and 0.25 m depth); Supplementary Fig. 6). The ubiquitous radiocarbon enrichment of deep peat porewater dissolved organic carbon, $CO_2$ and methane ($CH_4$) relative to

the surrounding solid peat is nonetheless clear evidence of dynamic solute transport at this site[13] and in other peatlands[20,43,50]. Shifts in water table position could lead to changes in porewater flow direction, but such shifts typically occur over longer time-scales in response to persistent changes in water table position[51,52]. It therefore appears unlikely that flow direction suddenly reverses every autumn at this location. We nonetheless recommend high-resolution tracer studies to disentangle the effect of increased turbulent diffusion together with dilution and potential shifts in mass flow during periods of weak thermal stability.

There was a poor coherence between increases in wind speed and the timing of the losses in porewater $CO_2$ store. Winds of high magnitude occurred across the whole year and were not limited to the periods of rapid losses in catotelm porewater $CO_2$ store (Supplementary Fig. 5). While wind is most likely an important source of kinetic energy in the peat porewater, its effect on turbulence is greatly modulated by the thermal stability. Furthermore, estimates of the air-water gas exchange coefficient ($k_{600}$) indicate low values, averaging 0.02 (SD ± 0.01) m d$^{-1}$ (Supplementary Fig. 4), suggesting that air-water gas exchange mostly operates within a limited section of the surficial porewater. This estimate of $k_{600}$ is on average one order of magnitude lower than those of small sheltered ponds, 0.19–0.72 m d$^{-1}$, possibly its closest analogue[53]. Therefore, even an increase in air-water gas exchange due to high winds alone could not strip $CO_2$ off the porewater at the depths observed in our data.

Lastly, ebullition is capable of rapidly mobilizing deep porewater gases directly to the atmosphere, a process that varies mostly with changes in atmospheric pressure and water table position[54–56]. Ebullition is most important for poorly soluble gases like $CH_4$. In comparison, $CO_2$ is about ten times more soluble than $CH_4$, and thus mostly found in the dissolved rather than free-phase. Ebullition is therefore a comparatively weaker transport process for $CO_2$ than $CH_4$. However, it is also worth considering that ebullition in the catotelm could represent an additional source of kinetic energy and turbulence in the porewater, thus indirectly enhancing the vertical diffusion of porewater $CO_2$ as a result. The various processes listed above certainly contribute to the variability in porewater $CO_2$ concentration, but none could single-handedly explain the magnitude, regularity or depth of the changes in porewater $CO_2$ store observed each autumn at this site.

**Implications for the peatland $CO_2$ budget**. The catotelm porewater $CO_2$ store has been treated as a temporally stable feature of peatland C budgets[12,15,16]. In contrast, our results demonstrate that this porewater $CO_2$ store (Fig. 6a) and the speed of vertical diffusion in the catotelm can vary widely across seasons (Fig. 4). The porewater $CO_2$ store in the top 2 m of the catotelm dropped by between 29 and 90 g C m$^{-2}$ during autumn when compared with the mean annual porewater $CO_2$ storage during periods with strong thermal stability (152 ± 6 g C m$^{-2}$) (Fig. 6a). These drops in catotelm porewater $CO_2$ store are considerable compared with other flux components of the peatland $CO_2$ budget, for example the net ecosystem exchange (NEE): −58 ± 21 g C m$^{-2}$ yr$^{-1}$ [57], and the annual stream $CO_2$ export 3 ± 0.75 g C m$^{-2}$ yr$^{-1}$ [58,59]. Losses in catotelm porewater $CO_2$ store each autumn could therefore potentially contribute to the seasonality in peatland $CO_2$ cycling.

We estimated that atmospheric emission from the catotelm porewater corresponded to an average flux of 0.3 g C m$^{-2}$ d$^{-1}$ across the year (i.e. based on the estimated average air-water gas exchange coefficient ($k_{600}$; average 0.02 m d$^{-1}$ (Supplementary Fig. 4)) and the average porewater $CO_2$ concentration at 0.13 m depth (24 mg C L$^{-1}$)). This average annual flux comprises 39% of

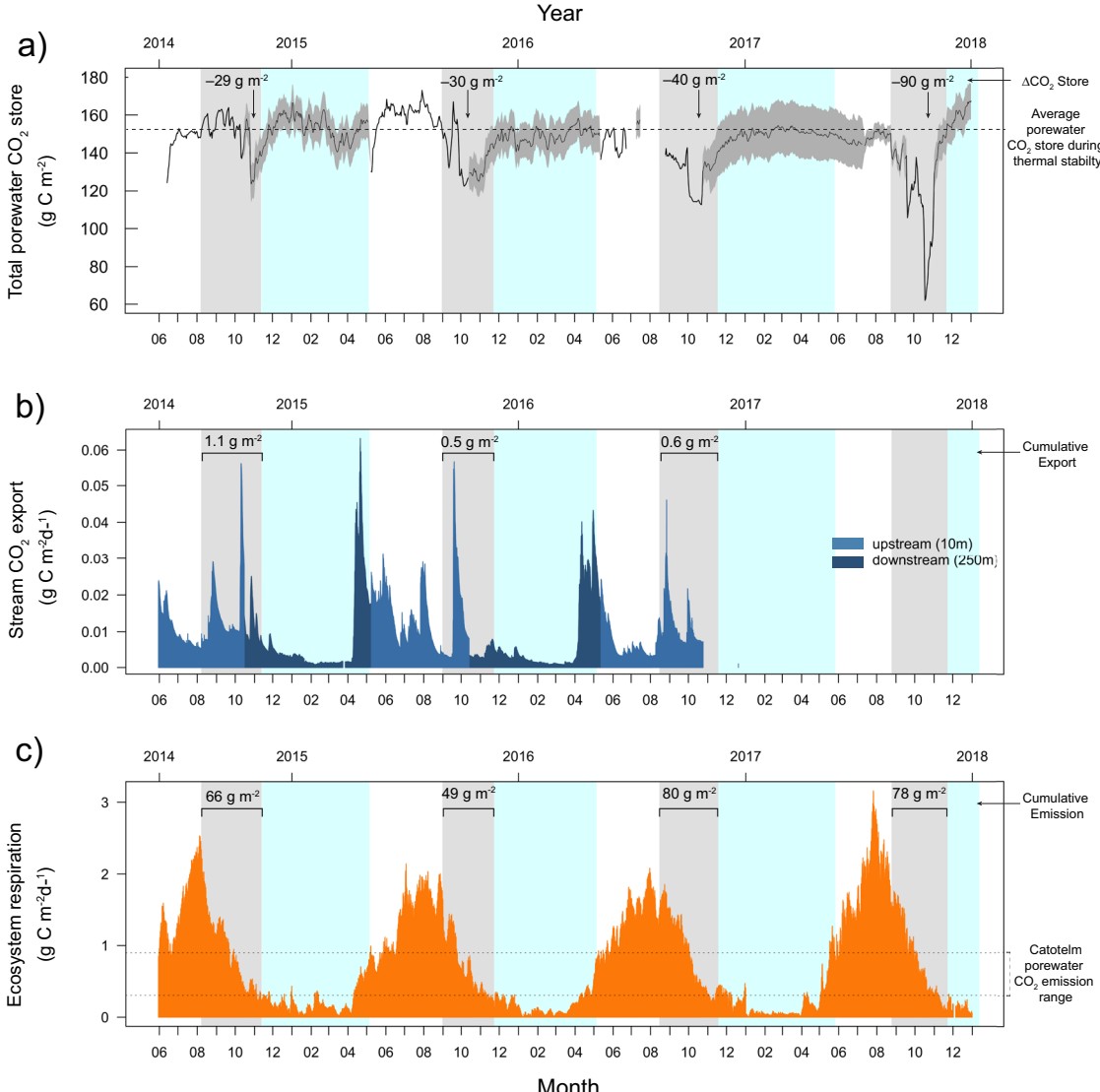

**Fig. 6 Catotelm porewater $CO_2$ store compared with atmospheric and hydrological $CO_2$ fluxes.** Timeseries of the **a** estimated catotelm porewater $CO_2$ store (g C m$^{-2}$) per unit land area in the top 2 m of the peat profile, **b** stream $CO_2$ export (g C m$^{-2}$) per unit land area at the upstream (light blue) and downstream (dark blue) location relative to the stream initiation point, and **c** the ecosystem respiration ($R_{eco}$, g C m$^{-2}$) per unit land area based on eddy-covariance measurements from June 2014 to January 2018. In **a**, the grey area shows the possible range of porewater $CO_2$ store for periods where measurements were unavailable in a certain layer while the black line shows the average porewater $CO_2$ store. In each panel, the x-axes indicate the dates, with years on top and month numbers on the bottom axis. The background areas, coloured in grey and cyan, mark the periods of weak thermal stability and ice/snow cover on the peatland surface, respectively.

the mean annual ecosystem respiration ($0.67 \pm 0.6$ g C m$^{-2}$ d$^{-1}$), indicating that $CO_2$ emissions from the catotelm porewater may control a significant share of the peatland-atmosphere $CO_2$ exchange dynamics throughout the year. Allowing for the rapid diffusion of deeper catotelm porewater $CO_2$ towards the surface, fostered by the sudden increase in turbulent diffusion in autumn, could increase this baseline flux by a factor of 3 ($0.9$ g C m$^{-2}$ d$^{-1}$) (i.e., assuming average $k_{600}$ conditions and surficial porewater $CO_2$ concentration corresponding to average concentrations observed at 0.75 m depth ($84$ mg C L$^{-1}$)). Whether the weakening of thermal stability, increased wind shear, changing water table position or changing vegetation cover, could further enhance the $k_{600}$ in autumn deserves closer examination. There was a steady decline in ecosystem respiration throughout autumn, but fluxes generally exceeded this range of catotelm porewater $CO_2$ emission (Figure 6c). There were also multiple peaks in ecosystem respiration each autumn, which could have occurred in

connection with the increase in turbulent diffusion during periods of weak thermal stability (Figure 6). The cumulative ecosystem respiration over the periods of weak thermal stability ($49$–$80$ g C m$^{-2}$ d$^{-1}$ across individual years (Fig. 6c)), was comparable to the periodic drops in porewater $CO_2$ store each autumn (Fig. 6a). A more detailed investigation of the interplay between changes in porewater $CO_2$ store and the peatland-atmospheric $CO_2$ exchange is recommended to further elucidate these aspects.

Hydrological export is another key removal process of porewater $CO_2$. The studied peat profile contains two layers of preferential lateral flow; the surficial peat and a deeper one located ~2 m below ground ($k_{sat}$ in Fig. 5)[13,58]. Porewater at both of these depths contains less $CO_2$ than the intermediate peat layers (0.5–1.5 m) (Fig. 1a). Rapid diffusion of porewater $CO_2$ towards these two layers of preferential flow could increase $CO_2$ export to the stream. Continuous measurements of stream $CO_2$ concentration during the

ice-free season recorded pulses of $CO_2$—rich water into the stream during autumn (e.g., October 2014, August and September 2015, and September and October 2016 (Supplementary Fig. 7). However, the mass of $CO_2$ exported to the stream outlet over the autumn appears negligible (0.5–1.1 g C m$^{-2}$ yr$^{-1}$) compared with the recurring losses from the porewater $CO_2$ store during autumn (Fig. 6b). Periods of weak porewater thermal stability contribute nonetheless 25–53% of the long-term annual stream $CO_2$ export ($3 \pm 0.7$ g C m$^{-2}$ yr$^{-1}$ [58]). Our studied peat profile is located in a zone of flow convergence and in close proximity to the stream initiation point (ca. 70-m distance), which could increase specific discharge at this site compared with other areas of the peatland. Areas of preferential flow are known for holding a disproportionate contribution to stream $CO_2$ export relative to other areas of the peatland[41,60,61]. We therefore consider that hydrological export to the stream could still explain a small part of the observed losses in catotelm porewater $CO_2$ store in autumn at this site. More detailed studies will be necessary to elucidate the full implications of changes in catotelm porewater $CO_2$ store for the peatland C budget.

The high-frequency observations of porewater $CO_2$ concentration provide evidence of regular destabilization of the catotelm porewater $CO_2$ store during periods of weakened porewater thermal stability that recur every autumn. To date, molecular diffusion was considered the main vertical transport pathway for catotelm porewater $CO_2$ towards the surface. Our analysis demonstrates that vertical diffusion of porewater $CO_2$ in the catotelm occurs at rates orders of magnitude greater than is possible by molecular diffusion. Vertical $CO_2$ diffusion occurs instead by turbulent diffusion, which is sensitive to changes in porewater thermal stability. This sensitivity can foster a sharp increase in porewater $CO_2$ diffusion from the catotelm to the surface, hence causing a recurring loss of catotelm porewater $CO_2$ store each autumn by atmospheric emission and hydrological export. We recommend further examination of the mechanism of turbulent diffusion in the catotelm of northern peatlands. Our findings reveal surprising dynamics in the catotelm porewater $CO_2$ store in a northern peatland. The implication of such dynamism for the peatland C budget, in particular for methane emissions to the atmosphere, have yet to be fully resolved. The catotelm porewater $CO_2$ store represents the equivalent of three consecutive years of net carbon accumulation by this peatland[59]. Thus, even slight fluctuations in the porewater $CO_2$ store could have significant implications for the peatland $CO_2$ sink capacity. The porewater of northern peatlands thus represents a dynamic, but possibly misunderstood component of peatland C cycling.

## Methods

**Study site**. This study was conducted at Degerö Stormyr, a 6.5 km$^2$ mire complex, located in Northern Sweden, at a topographic high point (~270 m.a.s.l.) about 60 km north-west of Umeå, Sweden (64°11′N, 19°33′E). The peatland is classified as an oligotrophic fen and is mostly undisturbed. Degerö Stormyr is composed of several inter-connected peatlands, separated by islets and ridges of glacial till soils. The study was conducted in a section draining 2.7 km$^2$ of the total peatland complex, which is dominated by the fen (70%), but contains forested areas on the outskirts of the catchment (30%). Forested areas are about one km horizontal distance away from the studied peat depth profile. The peat depth profile consists of accumulated peat in the top 3 m, which overlays a ~1-m-thick layer of ancient organic lake sediment, for a total depth of about 4 m. The bulk density of the peat depth profile in the top 2 m is low and averages $0.016 \pm 0.009$ g cm$^{-3}$, which corresponds to an average porosity of 98% (Supplementary Fig. 2). The peat depth profile is located ~70 m away from the initiation point of the stream outlet. The stream is found in an area of flow convergence where deep porewater is forced to the surface by a shallowing underlying mineral soil layer. The stream flow and C export are generated mostly through two conductive peat layers. The first consists of a preferential flow layer (i.e., deep macropore) at ~ 2 m below ground surface (Supplementary Fig. 2). The second is confined to the surface peat porewater [0–0.15 m below ground surface], with fluctuating contribution based on water table levels[13,58].

Degerö Stormyr is part of the European research infrastructure Integrated Carbon Observation System (ICOS) and Swedish Infrastructure for Ecosystem Science (SITES), through which the site has acquired a long historical record of atmospheric $CO_2$ and $CH_4$ exchange via eddy-covariance based measurements (since 2001)[57,59,62], hydrological C export[58], and meteorological observations. The climate in this region is cold temperate humid, with a 30 year (1981–2010) mean annual precipitation of 614 mm and mean annual temperature of 1.8 °C. Maximum average temperature typically occurs in July (14.7 °C), while the minimum average temperature is usually reached in January (−9.5 °C)[63]. The peatland bears a persistent snow and ice cover from November to early May, with the ice typically reaching down to 10–30 cm below ground surface[64]. The underlying geology comprises base-poor Svecofennian metasediments/metagreywacke (Geological Survey of Sweden, Uppsala, Sweden). The vegetation field layer is dominated by lawn and carpet plant communities dominated by *Eriophorum vaginatum* L., *Trichophorum cespitosum* (L.) Hartm., *Vaccinium oxycoccos* L., *Andromeda polifolia* L., and *Rubus chamaemorus* L., The bottom layer consists ~100% of Sphagnum mosses, dominated by *Sphagnum balticum* (Russ.) C. Jens., *Sphagnum lindbergii* Schimp. in Lindb., *Sphagnum majus* (Russ.) C. Jens. and *Sphagnum papillosum* Lindb. in the lawn and carpets with *Sphagnum fuscum* (Schimp.) Klinggr. dominating the sparse hummocks and ridges.

**Peat depth profile instrumentation**. A 4-m-deep peat profile was equipped with seven groundwater wells placed at different depths and screened for specific peat horizons via open slits along the wells ([0–0.25 m], [0.25–0.5 m], [0–0.5 m], [0.5–1 m], [1–2 m], [2–3 m], [3–4 m]). These wells were instrumented with $CO_2$ sensors and thermistors. An additional tube, 1-m-deep, was installed and equipped with a pressure transducer (MJK 1400, 0–1 m, MJK Automation AB) for water level measurements. The top of the wells was sealed with thick rubber bungs. The seal was essential to prevent atmospheric gas exchange, but could potentially affect the water exchange within the tube when the groundwater table varied. However, we consider that the close fit between the slitted walls of the tube and the sensor allowed gases from the surrounding peat to diffuse into the tube making porewater gas measurements accurate. The wells had a 31 mm inner diameter, which allowed for a close fit (6.5 mm gap) around the $CO_2$ sensors (18 mm diameter). The tubes were opened for sensor maintenance in May 2014 and 2015, to retrieve sensors in the shallow tubes in October of each year, and to redistribute sensors in August 2016 and September 2017.

Hourly measurements of the partial pressure of $CO_2$ ($pCO_2$) were made using the Vaisala CARBOCAP GMP221 nondispersive infrared (NDIR) $CO_2$ sensors (range 0–20%). These sensors have been evaluated in soils and surface waters spanning a wide range in temperatures and ambient pressures[65]. The $CO_2$ sensors were deployed at specific depths (0.13, 0.25, 0.38, 0.75, 1.5 m depth) in individual groundwater wells. Hourly $CO_2$ measurements at 2.5 m depth were conducted, but only between June 14 and August 8, 2014. The deepest porewater (2–4 m below ground) was sampled manually for $CO_2$ concentration on a monthly basis during the ice-free season in 2014 and 2015[13]. Sensor damage due to lightning strikes occasionally forced us to redistribute the sensors across the peat depth profile (e.g. measuring at 0.25 m instead of 0.13 and 0.38 m simultaneously). The $CO_2$ sensors within the top 0.5 m of the peat profile were removed between November and May to prevent frost damage.

Each sensor was enclosed inside a water-tight, gas-permeable Teflon membrane (PTFE) and sealed with Plasti Dip (Plasti Dip international, Baine, MN, USA) to ensure that the sensor was protected from water, but remained permeable to gases. The Teflon membranes were replaced in May 2015 and 2016, following ice-melt. Concentrations of $CO_2$ (expressed in mg C L$^{-1}$) were determined from the $pCO_2$ measurements considering water temperature (according to Henry's law), hydraulic and atmospheric pressure[65]. Hourly porewater temperature measurements were conducted along the full 4-m depth profile (0.13, 0.25, 0.38, 0.75, 1.5, 2.5, 3.5 m depth) in individual wells using thermistors (TO3R, TOJO Skogsteknik). All continuously measured data were stored on an external data logger (CR1000, Campbell Sci.).

**Porewater $CO_2$ store estimation**. The total porewater $CO_2$ store for the top 2 m of the peat depth profile was estimated by deriving the sum of the porewater $CO_2$ store for three individual porewater layers (i.e. [0 to 0.5 m], [0.5 to 1 m], [1 to 2 m]). The porewater $CO_2$ store for each porewater layer was calculated from the volume-weighted daily averaged $CO_2$ concentration at different depths (i.e. (summing the average of 0.13 m and 0.38 m or 0.25 m for [0 to 0.5 m], 0.75 m for [0.5 to 1 m], 1.5 m for [1 to 2 m]). The $CO_2$ store for each layer was adjusted for the porewater volume (average porosity 98%, Supplementary Fig. 2). For the surface porewater (0 and 0.5 m), the volume-weighting also included changes in water table position, in order to account for $CO_2$ dilution and concentration with rising and falling water table. When $CO_2$ concentration measurements were unavailable in a given layer (e.g. during winter between 0 to 0.5 m or periods following sensor damage), we applied the annual average $CO_2$ concentration for these specific depths, together with the minimum and maximum, to derive a continuous range of possible porewater $CO_2$ store.

**Thermal stability and turbulent diffusion.** The thermal stability of the porewater was estimated using the buoyancy frequency ($N^2$):

$$N^2 = -\frac{g}{\rho}\frac{d\rho}{dz} \qquad (2)$$

where $g$ is the gravitational acceleration (m s$^{-1}$), $\rho$ is the density of porewater at a given depth (kg m$^{-3}$), and $d\rho$ is the difference in density between the porewater at two different depths $dz$. The density of porewater was calculated based on water temperature assuming the absence of salinity. Periods of weak thermal stability were considered to begin when the difference in porewater temperature at 0.13 and 0.25 m was below 0.2 °C and ended when with freezing of the surface porewater (below 1 °C at 0.13 m). Periods of weak thermal stability therefore began in mid-August or early September, and ended in mid-November, lasting from 82 to 96 days across the 4 different years of observation. During periods of weak thermal stability, there were also weak inversions of the vertical porewater density gradient (i.e. slightly denser water above than below), which resulted in a negative vertical density gradient ($d\rho/dz$) and $N^2$. Under such conditions, it is possible for water in shallower peat to sink downwards because of gravity. However, we consider downward porewater flow (convection) in this peat profile to be small due to the sharp decrease of hydraulic conductivity of the peat with depth[38] (Supplementary Fig. 3) and the limited length of time during which those inversions occurred. Monthly measurements of porewater stable isotope ratio did not indicate large-scale downward fluid motion during autumn (Supplementary Fig. 6).

Turbulent diffusion describes the vertical transport of mass, heat and momentum induced by random fluid motion (eddies), and is usually several orders of magnitude greater than molecular diffusion. An apparent diffusion coefficient ($K_{app}$) can be derived using the vertically distributed peat porewater temperature time-series (i.e. heat budget method[26]). Vertical diffusive transport is considered to be turbulent when it exceeds the rate of molecular diffusion (i.e. $>10^{-7}$ m$^2$ s$^{-1}$). In the absence of light, and assuming negligible lateral heat transfer, the vertical transfer of heat from one layer to the layer below is driven by diffusion. Vertical diffusion at depth $z$ can be estimated as[27]:

$$K_{app} = \int_{max\ depth}^{z} A(z')\frac{\partial T(z')}{\partial t}dz'\left[A(z)\frac{\partial T(z)}{\partial z}\right]^{-1} \qquad (3)$$

where $K_{app}$ is the apparent diffusion coefficient (m$^2$ s$^{-1}$) and the temperature temporal gradient of the depths below $\frac{\partial T(z')}{\partial t}$, are calculated as the linear slope of temperature change over time, measured between seven days before and after the selected date (15 days in total), and $\frac{\partial T(z)}{\partial z}$ is the local ($z$) vertical daily average temperature gradient. $A$ is the 2D planar area at depth. Daily $K_{app}$ estimates were not considered when the vertical temperature gradient was no longer stratified, that is if the temperature slope has $r^2 < 0.7$ or is not statistically significant ($p > 0.05$), and/or when the vertical temperature gradient is inverted (i.e. colder temperatures above warmer temperatures). The coefficient of variation (C.V.) of the porewater $CO_2$ concentration was estimated over a 7-day moving window to assess the stability of porewater $CO_2$ concentration at each porewater depth.

**Atmospheric and hydrological $CO_2$ fluxes.** We investigated the potential implications for the peatland C budget of a rapid diffusion of catotelm porewater $CO_2$ towards the surface in autumn. We specifically quantified the poten tial emission of catotelm porewater $CO_2$ to the atmosphere and the hydrological $CO_2$ export to the stream outlet in the studied peatland. Atmospheric emission of dissolved porewater $CO_2$ can be described using Fick's first law, through which an estimate of the gas exchange coefficient ($k_{600}$) at the air–water interface can be derived:

$$k_C = F_C/(C_{pw} - C_{eq}) \qquad (4)$$

Where $F_C$ is the gas flux at the air–water interface, $k_C$ is the gas exchange coefficient (see Eq. (5) below), $C_{pw}$ is the gas concentration in the near-surface porewater and $C_{eq}$ is the concentration of gas in porewater at equilibrium with the atmosphere. This equation was used to estimate the $k_C$ between the surface porewater and the atmosphere. For $F_C$, we used continuous measurements of atmospheric $CH_4$ fluxes by eddy-covariance (ICOS; Degerö), since the peatland-atmosphere $CH_4$ exchange is unidirectional (upward) and ascribed mostly to the anoxic water-saturated peat. In comparison, the atmospheric $CO_2$ fluxes are bi-directional (both downward and upward) and largely influenced by primary production above ground making them unsuitable for estimating the $k_C$. Steady ebulitive $CH_4$ fluxes measured by the eddy-covariance flux tower, could lead to an overestimation of the $k_C$, but this pathway of $CH_4$ release is typically episodic[54,56], thus mostly causing isolated over-estimations. The range in $C_{pw}$ was determined from the mean, maximum and minimum porewater $pCH_4$ at 0.13 m depth[66], while the $C_{eq}$ was set at 1.7 ppm. Ambient $C_{pw}$ and $C_{eq}$ concentrations were determined according to porewater temperature (Henry's law), hydraulic and atmospheric pressure. The $k_C$ was standardised for $CO_2$ at 20 °C ($k_{600}$) based on the following equation:

$$k_{600} = k_C(600/Sc_{CH4})^{-2/3} \qquad (5)$$

Where $Sc_{CH4}$ is the Schmidt number based on the surface porewater temperature[67]. The eddy-covariance flux tower, where peatland-atmosphere $CO_2$

and $CH_4$ exchange is measured, is located on the same peatland but ~1 km away from the peat profile measurements, which could lead to important differences in C-exchange dynamics. However, both areas are characterized by a similar micro-topographical relief vegetation. The $k_{600}$ estimate allowed us to determine an annual average rate of $CO_2$ emission from the porewater to the atmosphere and estimate its possible increase due to the rise of turbulent diffusion in the catotelm porewater during autumn.

The hydrological $CO_2$ export was estimated from the combined continuous stream $CO_2$ concentration and discharge measurements, standardized by the estimated catchment area (2.7 km$^2$). The stream $CO_2$ concentration measurements were carried out using the same $CO_2$ sensor methodology as used for the peat porewater described above. The sensors were deployed about 10 and 250 m downstream of the stream initiation point[13]. The stream $CO_2$ concentration measurements in the upstream location were performed only during the ice-free season, but year round measurements were carried out in the downstream location. Stream $CO_2$ export was estimated at both locations in order to derive a more complete estimate over time. The stream discharge was determined by applying a stage height-discharge rating curve to hourly water level measurements. Stream water height measurements were conducted throughout the year at a 10 m long trapezoidal flume inside a heated dam house ~50 m downstream of the stream initiation point. All calculations and analyses were performed using R (R Core Team, 2021)[68].

## Data availability

The hourly measurements of porewater $CO_2$ concentration, temperature and water table position, and estimates of daily average porewater $CO_2$ concentration, coefficient of variation of porewater $CO_2$ concentration, total porewater $CO_2$ store and apparent diffusion coefficient, have been deposited in the Swedish National Data Service [https://doi.org/10.5878/ggdt-ew12].

## Code availability

R codes related to this paper are available from the corresponding author upon request.

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

## Acknowledgements

This study was supported by the Swedish Research Council (grant #2012-3919 to K.B.), the Department of Earth Sciences at Uppsala University, and FORMAS a Swedish Research Council for Sustainable Development (grant #2019-01592 to A.C.). We thank the Krycklan

Catchment Study crew for field support. We also thank Jelmer Nijp for providing measurements of the saturated peat horizontal hydraulic conductivity. We further thank Andy Baird for constructive input during the peer review of this publication, as well as Nigel Roulet and M.Bayani Cardenas for inputs at an earlier stage of the study. Additional support by the two research infrastructures ICOS (Integrated Carbon Observatory System) Sweden and SITES (Swedish Infrastructure for Ecosystem Science), both funded by the Swedish Research Council and participating institutes, are gratefully acknowledged.

## Author contributions

A.C., M.B.W. and K.B. designed the study. M.B.W. and K.B. contributed instrumentation and funding, and M.N. provided infrastructure for the data collection. A.C. carried out the fieldwork, processed and analysed the data. A.C., D.V. and M.B.W. wrote the paper. M.N. and K.B. provided scientific insights to the analysis and interpretation of the data. All authors commented on earlier versions of this paper.

## Funding

## Competing interests

The authors declare no competing interests.
