## [Peer Review File · Nature Communications]

REVIEWER COMMENTS

Reviewer #1 (Remarks to the Author):

Summary: This paper describes thermal destabilization in a peatland which the authors suggest contributes to increasing flux of CO₂ from deep peat porewater in the autumn. The high rate of flux is comparable to the C sink of this peatland suggesting a mechanism for a potentially strong peatland-climate feedback.

General Impressions: I generally like the idea of thermal destabilization as a mechanism for gas transport in peatlands. I think the authors could consult more of the recent peatland literature to better contextualize their study and results. The methods lack clarity in a few points specifically with regards to the flux calculations (detailed comments below) that I think needs to be revised before the study results can be evaluated fully.

1. I am interested in the idea of rapid diffusion of CO₂ facilitated by thermal instability, but I'm not convinced about the opening premise that deep CO₂ in peatlands is generally considered to be inert. Several of the modeling groups explicitly model diffusion and ebullition of CO₂ from deep towards the surface (e.g. Ma et al., 2017).

Ma, S., Jiang, J., Huang, Y., Shi, Z., Wilson, R.M., Ricciuto, D., Sebestyen, S.D., Hanson, P.J. and Luo, Y., 2017. Data-constrained projections of methane fluxes in a northern Minnesota peatland in response to elevated CO₂ and warming. *Journal of Geophysical Research: Biogeosciences*, 122(11), pp.2841-2861.

2. Line 30-31: The authors state that the ability of peatlands to act as a sink is attributable to processes in the oxic surface. I would argue that is not the case and the sink ability of peatlands occurs when peat is buried in the catotelm away from energetic TEAs and protected from high rates of microbial decomposition by numerous processes including cold, low pH, and accumulation of potentially inhibitory compounds. The ability of peatlands to act as a sink depends on the balance between primary production forming peat and the ability of microorganisms to remineralize. If PP is high relative to decomposition, the peatland is a sink, if PP is low relative to decomposition then they act as a source.

3. Line 35-36: I think this was the predominant thinking years ago, but it has since been shown that peatlands may be hydrologically active much deeper in the profile. See work by Paul Glaser, Jeff Chanton etc.

4. Line 37-39 This statement needs some references.

5. Par 41-55: What about ebullition? You don't need saturated concentrations of methane to form bubbles (see Chanton and Dacey 1991 and Chanton and Whiting 1995)

Chanton, J.P. and Dacey, J.W.H., 1991. Effects of vegetation on methane flux, reservoirs, and carbon isotopic composition, In "Trace Gas Emissions by Plants", edited by TD Sharkey, EA Holland, and HA Mooney.

Chanton, J.P. and Whiting, G.J., 1995. Trace gas exchange in freshwater and coastal marine environments: ebullition and transport by plants. *Biogenic trace gases: measuring emissions from soil and water*, pp.98-125.

6. Line 57-59: While Clymo did say this approx. a quarter century ago, more recent studies challenge this assertion. (see work by Paul Glaser).

7. Line 60: what is meant by "manual characterization" exactly?

8. General intro: Although one of my papers was cited as supporting evidence, I wouldn't say that I believe deep porewater CO₂ is "static".

9. Lines 203-206: it seems contradictory to me to talk about a molecule of CO₂ taking 1.5 years to reach the surface during one season. Although I think I know what the authors mean, it is just confusing. I think this would be better expressed as a comparison of velocities.

10. Line 227: simultaneous

11. Line 241-242: This statement is incorrect, please see Kolton et al., 2019. Although their work showed a low T max in CO₂ production there was a precipitous drop-off just over 5°C.

Kolton, M., Marks, A., Wilson, R.M., Chanton, J.P. and Kostka, J.E., 2019. Impact of warming on greenhouse gas production and microbial diversity in anoxic peat from a Sphagnum-dominated bog (Grand Rapids, Minnesota, United States). *Frontiers in microbiology*, 10, p.870.

12. Line 297-300: Something to note is that the water table in Sphagnum-dominated peatlands can be below the surficial sphagnum layer which may reduce wind-shear air-water exchange relative to the small ponds mentioned.

13. Line 327-329: I think this calculation needs to take into account the approximately 10 times higher solubility of CO₂ relative to CH₄ in these conditions so CO₂ is likely much lower than predicted here.
14. Line 439: should this be >2meters?
15. Line 440-442: I don't understand this. The deepest porewater (2m) was sampled once a month, but 2.5m was sampled hourly in some months? This part needs clarification.
16. Paragraphs 522-552: I've done some diffusion modeling and I can't quite follow what they are doing here. For example, Is the mass transfer method (e.g. Happell et al., 1995) or the boundary layer method used to calculate exchange coefficient here? If the latter, the result is very sensitive to boundary layer thickness. How was that chosen? I think this section could use some clarification. (see: MacIntyre, S., Wanninkhof, R. and Chanton, J.P., 1995. Trace gas exchange across the air-water interface in freshwater and coastal marine environments, in, edited by: Matson, PA and Harriss, RC, Biogenic trace gases: Measuring emissions from soil and water. Methods in ecology, 52-97. Happell, J.D., Chanton, J.P. and Showers, W.J., 1995. Methane transfer across the water-air interface in stagnant wooded swamps of Florida: Evaluation of mass-transfer coefficients and isotropic fractionation. Limnology and Oceanography, 40(2), pp.290-298.

Reviewer #2 (Remarks to the Author):

The manuscript presents evidence of a possible source of carbon dioxide losses from peatlands that has been ignored previously. The authors argue that turbulent diffusion has the potential to move deep porewater CO₂ closer to the peatland surface, where it could be released by atmospheric emission and hydrological export. Their estimates of autumn loss of porewater CO₂ storage for a Swedish peatland are comparable to the peatland's annual C sink. The authors challenge the perception that greenhouse gases in deep peat porewater are largely inert and raise the possibility of substantial errors in global estimates of peatland C sinks and GHG emission. The results are original and should be of interest to peatland researchers as well as to researchers with wider interests in the terrestrial carbon cycle and greenhouse gas emissions.

The key argument is supported by analysis of four years of near-continuous measurements of porewater CO₂ and temperature at varying depths in a peatland in northern Sweden. The authors used a heat budget approach to estimate the turbulent diffusion coefficient for CO₂. The temporal extent and resolution of porewater CO₂ data are rare, if not unique to this study. The sampling strategy is valid for demonstrating recurring thermal instabilities and associated seasonal fluctuations in porewater CO₂ concentrations and storage. Previous studies based on mineral solute profiles (e.g., Griffiths and Sebestyen 2016) have already demonstrated that deep porewater is dynamic rather than static. The unique contribution of this study is to demonstrate this phenomenon for dissolved CO₂ and to propose that these seasonal porewater dynamics are explained by turbulent diffusion.

The section of the manuscript that considers the implications of turbulent diffusion for the peatland C budget is less convincing. The authors argue that rapid rise of deep porewater to the surface could release CO₂ to the atmosphere and/or to runoff. They estimated the cumulative porewater CO₂ loss through autumn as comparable to the peatland's annual net ecosystem exchange and an order of magnitude larger than its stream CO₂ export, and the daily flux of deep porewater CO₂ toward the surface during periods of instability as comparable to mean annual rate of ecosystem respiration. The difficulty is that the measurements of atmospheric net CO₂ exchange and stream CO₂ export do not show the pattern or magnitude of fluxes that would be expected. To account for this lack of evidence for rapid efflux of CO₂, the authors argue that the method of partitioning NEE into GPP and Reco is flawed. Further, they argue that their peat depth profile may be unrepresentative of the whole catchment, explaining why autumn pulses of stream CO₂ export were negligible compared to losses of porewater CO₂ storage.

The possibility of substantial losses of deep porewater CO₂ is tantalising, but the evidence for its rapid efflux to the atmosphere or stream water is weak: based on the data available, the porewater CO₂ budget is

incomplete. It would be helpful to outline what measurements and experimental set-ups would be required to confirm and quantify C losses from deep porewater. For example, peat profile monitoring of hydraulic heads and the concentrations of a conservative solute would help to disentangle the contributions to vertical mass transport of advection, molecular diffusion and turbulent diffusion. Tracer studies would help to establish the fate of 'lost' porewater CO₂ and, ideally, CH₄.

Reference

Griffiths, N.A., and Sebestyen, S.D. (2016) Dynamic Vertical Profiles of Peat Porewater Chemistry in a Northern Peatland. *Wetlands* 36:1119–1130. DOI 10.1007/s13157-016-0829-5

Specific comments:

Line 36 – reference needed for 'hydro-physically inactive'

Line 41 – Does 'CO₂' refer to concentration or stock? Spelling: 'instantaneous'

Line 60 – 'manual characterization with a low time resolution' – rephrase for clarity, e.g., periodic manual measurements?

Line 111 (and following) – 'overlying'

Lines 120-121 – Is the second value for the 1-2 m layer (not 1.5 m)?

Lines 124-126 - Near-surface porewater CO₂ concentrations will also be affected by rhizosphere processes, which change seasonally.

Line 195 / Figure 5b – Given the relationship stated in the equation, it would make more sense for K_z and the trend lines to be plotted against 1/N₂ rather than N₂.

Line 206 – Spelling: example

Line 227 – Spelling: simultaneous

Line 285 – Spelling: temporarily

Line 328 – Spelling: simultaneous

Line 426 – Spelling: equipped

Line 431 – Spelling: bungs

Lines 467-468 – 'in order to account for dilution...'

Line 476 – 'volumetric unit area' – unclear what this means, as the dimensions are [mass] [volume]⁻¹ [time]⁻¹

Line 507 – Does this mean 15 days in total?

Line 537 – Spelling: dynamics

Reviewer #3 (Remarks to the Author):

Review of Campeau et al. 'Autumn destabilization of deep porewater CO₂ storage in a northern peatland driven by turbulent diffusion' submitted to *Nature Communications*.

Overview

This paper is a thought-provoking account of a high-quality dataset from a northern peatland that suggests that CO₂ dissolved in deep (> 0.5-0.75 m depth) porewater can be lost from peatlands during the autumn. Previously, this store of CO₂ has been assumed to be largely static or having a slow turnover. In a mostly well-written paper, the authors suggest that their dissolved CO₂ data can be explained by the process of turbulent diffusion, which can lead to rates of dissolved CO₂ transport that are orders of magnitude greater than molecular transport. In a careful analysis, the authors also consider the importance of the posited process compared to other processes in the carbon cycle in peatlands. Overall, I enjoyed reading the paper and I believe there is information in it that is very worthy of publication in *Nature Communications*. However, I struggled to understand parts of the manuscript and I believe the authors could do a better job of explaining their proposed mechanism. Below, I explain how I think the paper could be improved. I have also marked up the pdfs of the manuscript and the SI, and the editor and authors are referred to these for other comments and concerns that I recommend are addressed.

More than one type of turbulent diffusion?

Groundwater scientists usually consider there to be two ways by which solute mixing may occur: molecular

diffusion and mechanical dispersion, the latter being analogous to diffusion but occurring in laminar flows due to the 'splitting' of flow through a porous medium as flowing water 'samples' the wide range of pore sizes. Turbulence can sometimes occur in porous media in, for example, biogenic macropores such as wormholes, larger voids between soil peds, and joints in bedrock. Turbulent transfer of a range of properties is also known to occur in surface waters, and the transfer has been found to follow mathematically the diffusion equation; hence, the use of the term 'turbulent diffusion'. The latter occurs when there is rapid mass transfer of water. The authors, however, seem to be suggesting a type of turbulent transfer that does not involve mass flow of water. From the text and Figure 6 they propose that pore-scale turbulent eddies form that help transfer dissolved CO₂ along a concentration gradient. They also note that this turbulence causes a transfer of CO₂ that is orders of magnitude greater than would occur via molecular random walks.

In principle, I find this suggestion plausible, but what I find confusing is the authors' explanation, or lack of explanation, of the mechanisms involved. In short: what causes the turbulence? The authors are clear in suggesting mass flow is not responsible (e.g., on lines 226-7), and they warn against treating the pore-water domain in the same way as lake water where turbulent diffusion has also been observed. Perhaps it is my ignorance showing here, and perhaps other scientists will clearly understand what the authors mean. However, I suspect many target readers of the paper will be like me and not know how turbulent diffusion can occur without mass flow. I guess part of my problem here is that I struggle to see how turbulent mixing at the scale posited by the authors can occur in the pores found in the deeper peat at their site. They suggest that turbulent eddies in the peat pores might form at scales of mm to cm but I don't see how that is possible because most pores in the deep peat will be of mm scale or much less (based on the hydraulic conductivity data presented in the SI). Notably, on lines 227-235, the authors themselves seem somewhat uncertain about how turbulent diffusion can occur in peat in the absence of mass flow.

Ebullition and measurement of dissolved CO₂

I wonder too whether bubble formation and transport within the peat might also offer a part explanation of the authors' findings. Biogenic gas bubbles tend to accumulate in peat during the growing season and reach a peak unit volume by late summer and early autumn. These bubbles contain CO₂ as well as CH₄ and their movement upwards through the peat profile may also facilitate CO₂ transfer. Their movement through, and loss from, the peat profile will also cause mixing of pore water that might give rise to turbulent diffusive transfer of dissolved CO₂. I should add that sudden episodes of gas bubble loss from peat are not always easily detected or measured by eddy correlation. The flux tower used by the authors was also somewhat distant from the monitored peat profile so may have missed ebullition events near the sensors.

I have a slight concern about the measurement of dissolved CO₂. What happens to the CO₂ sensors if bubbles are in contact with the sensor membrane? Is it possible that the sensors measure a combination of dissolved and free-phase CO₂? Also, if bungs were fitted to the tops of the sampling tubes, how did flow of water into and out of the tubes occur? Such tubes need to be vented to allow water ingress and egress.

Recommendation

Overall, I recommend the paper is revised and reviewed again. I think the authors need to explain their proposed mechanism more clearly, so that wider readership can understand it. Currently, I have some doubts about what the authors are proposing, and I'd like to be more convinced that their mechanism is sound.

Please note that I operate a policy of 'open reviewing'. My comments above, and on the marked-up manuscript and SI, are for the editor and the authors and I would like my identity to be revealed to the authors.

Andy Baird,
Chair of Wetland Science
University of Leeds, UK;
5th August 2021.

Response to Reviewers

Reviewer #1 (Remarks to the Author):

Summary: This paper describes thermal destabilization in a peatland which the authors suggest contributes to increasing flux of CO₂ from deep peat porewater in the autumn. The high rate of flux is comparable to the C sink of this peatland suggesting a mechanism for a potentially strong peatland-climate feedback.

General Impressions: I generally like the idea of thermal destabilization as a mechanism for gas transport in peatlands. I think the authors could consult more of the recent peatland literature to better contextualize their study and results. The methods lack clarity in a few points specifically with regards to the flux calculations (detailed comments below) that I think needs to be revised before the study results can be evaluated fully.

1. I am interested in the idea of rapid diffusion of CO₂ facilitated by thermal instability, but I'm not convinced about the opening premise that deep CO₂ in peatlands is generally considered to be inert. Several of the modelling groups explicitly model diffusion and ebullition of CO₂ from deep towards the surface (e.g., Ma et al., 2017). *Ma, S., Jiang, J., Huang, Y., Shi, Z., Wilson, R.M., Ricciuto, D., Sebestyen, S.D., Hanson, P.J. and Luo, Y., 2017. Data-constrained projections of methane fluxes in a northern Minnesota peatland in response to elevated CO₂ and warming. Journal of Geophysical Research: Biogeosciences, 122(11), pp.2841-2861*

We agree with the reviewer that this was an overstatement. Many studies have documented considerable dynamics in groundwater flow, with significant implications for peat solute and gas concentrations. The peat porewater CH₄ store, in particular, is known to be dynamic due to the role of ebullition. We nonetheless note a general agreement in the literature that vertical transport of porewater CO₂ (a more soluble gas than CH₄) is governed by molecular diffusion, which is a slow and constant removal process (Line 46-49).

We have revised the introduction to instead lay the concept of acrotelm and catotelm as two main layers of a peatlands, and detail their roles in peatland CO₂ cycle (atmospheric and hydrological fluxes) (Line 28-44). We avoid stating that the catotelm is inert, and rather detail its role in the peatland CO₂ cycling (which is lower than that of the acrotelm). We subsequently state the vertical transport of catotelm porewater CO₂ towards the acrotelm is considered to take place by molecular diffusion, a slow and constant process (Line 34-35 & 43-44), where we also refer to the modelling studies that you recommended (Ma et al, 2017) and (Walter et al, 2000). The previous version of the introduction listed multiple transport pathways (mass flow, diffusion, ebullition) for both CO₂ and CH₄ in peat porewater, which we consider was out of focus. Our study specifically challenges the assumption of a slow and constant rate of vertical diffusion in the catotelm, not the effect of other processes, which have been studied in more detail. These other hydrological processes, and their possible role in the peat porewater CO₂ dynamics, are instead detailed and discussed in the results & discussion section (e.g., Line 186-204, and section "Other drivers of porewater CO₂ dynamics (Line 221-270").

2. Line 30-31: The authors state that the ability of peatlands to act as a sink is attributable to processes in the oxic surface. I would argue that is not the case and the sink ability of peatlands occurs when peat is buried in the catotelm away from energetic TEAs and protected from high rates of microbial decomposition by numerous processes including cold, low pH, and accumulation of potentially inhibitory compounds. The ability of peatlands to act as a sink depends on the balance between primary production forming peat and the ability of microorganisms to remineralize. If PP is high relative to decomposition, the peatland is a sink, if PP is low relative to decomposition, then they act as a source.

This statement has been adjusted (Line 30-32).

3. Line 35-36: I think this was the predominant thinking years ago, but it has since been shown that peatlands may be hydrologically active much deeper in the profile. See work by Paul Glaser, Jeff Chanton etc. Reference to be added: Levy,

Z. F., D. I. Siegel, S. S. Dasgupta, P. H. Glaser, and J. M. Welker (2014), Stable isotopes of water show deep seasonal recharge in northern bogs and fens, *Hydrological Processes*, 28(18), 4938-4952, doi:10.1002/hyp.9983.

We now present a more detailed evaluation of the potential role of changes in mass flow to explain the losses of porewater CO₂ in autumn (Line 186-204 & 232-251), where we also cite the references, you recommended.

4. Line 37-39 This statement needs some references.

This statement has been removed because we judged it was not relevant to introduce the study.

5. Par 41-55: What about ebullition? You don't need saturated concentrations of methane to form bubbles (see Chanton and Dacey 1991 and Chanton and Whiting 1995) *Chanton, J.P. and Dacey, J.W.H., 1991. Effects of vegetation on methane flux, reservoirs, and carbon isotopic composition, In "Trace Gas Emissions by Plants", edited by TD Sharkey, EA Holland, and HA Mooney.*

Chanton, J.P. and Whiting, G.J., 1995. Trace gas exchange in freshwater and coastal marine environments: ebullition and transport by plants. Biogenic trace gases: measuring emissions from soil and water, pp.98-125.

We have removed mentions of ebullition in the introduction since this transport pathway is mostly relevant for CH₄ and less for CO₂ (more soluble). We nonetheless evaluate the potential role of ebullition in the regular losses of catotelm porewater CO₂ in autumn in the discussion (Line 261-268).

6. Line 57-59: While Clymo did say this approx. a quarter century ago, more recent studies challenge this assertion. (see work by Paul Glaser).

Now state that peatlands are traditionally viewed as two-layer systems (catotelm and acrotelm) and explain their respective roles in the peatland CO₂ cycling (Line 28-44).

7. Line 60: what is meant by "manual characterization" exactly?

We now specify that the measurements are carried under low temporal sampling resolution (Line 53-54).

8. General intro: Although one of my papers was cited as supporting evidence, I wouldn't say that I believe deep porewater CO₂ is "static".

This statement has been removed (see reply to comment #1).

9. Lines 203-206: it seems contradictory to me to talk about a molecule of CO₂ taking 1.5 years to reach the surface during one season. Although I think I know what the authors mean, it is just confusing. I think this would be better expressed as a comparison of velocities.

We now provide both speed and time estimate in this statement (Line 162-168). We consider that stating the transport rate in time units provides a more tangible unit for less familiar readers. Stating the time units is also informative to understand the rate of diffusion in relation with the sampling interval of porewater CO₂ concentration (i.e., hourly measurements).

10. Line 227: simultaneous

corrected

11. Line 241-242: This statement is incorrect, please see Kolton et al., 2019. Although their work showed a low T max in CO₂ production there was a precipitous drop-off just over 5°C.

Kolton, M., Marks, A., Wilson, R.M., Chanton, J.P. and Kostka, J.E., 2019. Impact of warming on greenhouse gas production and microbial diversity in anoxic peat from a Sphagnum-dominated bog (Grand Rapids, Minnesota, United States). Frontiers in microbiology, 10, p.870.

We thank the reviewer for providing this piece of information. We now provide more citations to this statement and include the study from Kolton et al 2019. However, our understanding of their results is that they report a peak in CO₂ production at 5°C. This helps reinforce our argument that a slowdown in respiration is unlikely the main driver of the phenomenon observed in our data (Line 226-230).

12. Line 297-300: Something to note is that the water table in Sphagnum-dominated peatlands can be below the surficial sphagnum layer which may reduce wind-shear air-water exchange relative to the small ponds mentioned.

We now explicitly mention vegetation cover as a potential source of variability in the air-water gas exchange (Line 294-295)

13. Line 327-329: I think this calculation needs to take into account the approximately 10 times higher solubility of CO₂ relative to CH₄ in these conditions so CO₂ is likely much lower than predicted here.

This statement has been removed. We now provide more detailed calculations of the possible rate of porewater CO₂ emission to the atmosphere (See comments from reviewer 2), based on the estimated air-water gas exchange coefficient (Methods Line 496-524), and porewater CO₂ concentrations (Discussion, Line 284-304).

All measurements of porewater CO₂ and CH₄ partial pressure are converted to concentrations based on Henry's law (according to ambient temperature and pressure) (Methods Line 432-433 & 511-513).

The differences in solubility between CO₂ and CH₄ is also mentioned in the text (Discussion Line 263-265).

14. Line 439: should this be >2meters?

Statement clarified (Line 422-424)

15. Line 440-442: I don't understand this. The deepest porewater (2m) was sampled once a month, but 2.5m was sampled hourly in some months? This part needs clarification.

Statement clarified (Line 422-424)

16. Paragraphs 522-552: I've done some diffusion modelling and I can't quite follow what they are doing here. For example, Is the mass transfer method (e.g., Happell et al., 1995) or the boundary layer method used to calculate exchange coefficient here? If the latter, the result is very sensitive to boundary layer thickness. How was that chosen? I think this section could use some clarification.

(see: MacIntyre, S., Wanninkhof, R. and Chanton, J.P., 1995. Trace gas exchange across the air-water interface in freshwater and coastal marine environments, in, edited by: Matson, PA and Harriss, RC, Biogenic trace gases: Measuring emissions from soil and water. Methods in ecology, 52–97. Happell, J.D., Chanton, J.P. and Showers, W.J., 1995. Methane transfer across the water-air interface in stagnant wooded swamps of Florida: Evaluation of mass transfer coefficients and isotopic fractionation. Limnology and Oceanography, 40(2), pp.290-298.

We have clarified this section of the methods (Line 496-524). We used the mass transfer method dictated by Fick's first law of diffusion (Eq. 1 in Happell et al. 1995 paper). Having continuous atmospheric CH₄ flux measurements and the measurements of porewater CH₄ concentration, we were able to estimate the air-water gas exchange coefficient (k_{600}).

We are aware of the uncertainties of our k_{600} estimates (Line 294-296), but we still consider that these estimates are useful in determining the possible rate of increase in porewater CO₂ emission to the atmosphere (Line 521-524), and the depth range of air-water gas exchange (Line 254-256). The estimated k_{600} values themselves are low and consistent with the environmental conditions and literature (Line 256-258).

We have recently deployed a CH₄ sensor in the surface peat porewater near the eddy covariance tower and intend to use these measurements in a future study to model the air-water gas exchange coefficient in peat porewater with changing water table position and wind shear.

Reviewer #2 (Remarks to the Author):

The manuscript presents evidence of a possible source of carbon dioxide losses from peatlands that has been ignored previously. The authors argue that turbulent diffusion has the potential to move deep porewater CO₂ closer to the peatland surface, where it could be released by atmospheric emission and hydrological export. Their estimates of autumn loss of porewater CO₂ storage for a Swedish peatland are comparable to the peatland's annual C sink. The authors challenge the perception that greenhouse gases in deep peat porewater are largely inert and raise the possibility of substantial errors in global estimates of peatland C sinks and GHG emission. The results are original and should be of interest to peatland researchers as well as to researchers with wider interests in the terrestrial carbon cycle and greenhouse gas emissions.

The key argument is supported by analysis of four years of near-continuous measurements of porewater CO₂ and temperature at varying depths in a peatland in northern Sweden. The authors used a heat budget approach to estimate the turbulent diffusion coefficient for CO₂. The temporal extent and resolution of porewater CO₂ data are rare, if not unique to this study. The sampling strategy is valid for demonstrating recurring thermal instabilities and associated seasonal fluctuations in porewater CO₂ concentrations and storage. Previous studies based on mineral solute profiles (e.g., Griffiths and Sebestyen 2016) have already demonstrated that deep porewater is dynamic rather than static. The unique contribution of this study is to demonstrate this phenomenon for dissolved CO₂ and to propose that these seasonal porewater dynamics are explained by turbulent diffusion.

The section of the manuscript that considers the implications of turbulent diffusion for the peatland C budget is less convincing. The authors argue that rapid rise of deep porewater to the surface could release CO₂ to the atmosphere and/or to runoff. They estimated the cumulative porewater CO₂ loss through autumn as comparable to the peatland's annual net ecosystem exchange and an order of magnitude larger than its stream CO₂ export, and the daily flux of deep porewater CO₂ toward the surface during periods of instability as comparable to mean annual rate of ecosystem respiration. The difficulty is that the measurements of atmospheric net CO₂ exchange and stream CO₂ export do not show the pattern or magnitude of fluxes that would be expected. To account for this lack of evidence for rapid efflux of CO₂, the authors argue that the method of partitioning NEE into GPP and Reco is flawed. Further, they argue that their peat depth profile may be unrepresentative of the whole catchment, explaining why autumn pulses of stream CO₂ export were negligible compared to losses of porewater CO₂ storage.

The possibility of substantial losses of deep porewater CO₂ is tantalizing, but the evidence for its rapid efflux to the atmosphere or stream water is weak: based on the data available, the porewater CO₂ budget is incomplete. It would be helpful to outline what measurements and experimental set-ups would be required to confirm and quantify C losses from deep porewater. For example, peat profile monitoring of hydraulic heads and the concentrations of a conservative solute would help to disentangle the contributions to vertical mass transport of advection, molecular diffusion and turbulent diffusion. Tracer studies would help to establish the fate of 'lost' porewater CO₂ and, ideally, CH₄.

Reference

Griffiths, N.A., and Sebestyen, S.D. (2016) Dynamic Vertical Profiles of Peat Porewater Chemistry in a Northern Peatland. *Wetlands* 36:1119–1130. DOI 10.1007/s13157-016-0829-5

We agree with the reviewer and thank him/her for pointing out this issue. The previous assessment of the possible release of catotelm porewater CO₂ towards the atmosphere or the stream outlet was indeed weak. We have revised these calculations and now provide a more thorough assessment of the potential increase in porewater CO₂ emission to the atmosphere and hydrological export.

The previous manuscript used the time-series of porewater CO₂ concentration to estimate apparent fluxes (i.e., based on daily changes at different depths). We consider that the resolution of our measurements across the depth profile was too low to generate reasonable estimates of the fluxes. For example, measurements at 1.5m were assumed to represent the full depth between 1m to 2m. These assumptions probably amplified the rate of porewater CO₂ change per volumetric unit area. We removed those flux estimates, but nonetheless present an estimate of the continuous total porewater CO₂ store, which we use to determine the magnitude of the regular losses in porewater CO₂ store (Figure 6a, Methods Line 440-451).

We now estimate the average annual CO₂ emission from the porewater to the atmosphere, based on K_{600} estimates (Method Line 493-524) and porewater gas concentrations, and determine the possible increase in atmospheric fluxes with a rise in porewater CO₂ concentration in the surficial porewater (Discussion Line 284-304). We then compare this range in porewater CO₂ emission with the time-series of ecosystem respiration (Line 296-297). This comparison reveals that ecosystem respiration rates in autumn generally exceeds this estimated range of porewater CO₂ emission (Figure 6c). This suggest that changes in porewater CO₂ emission could be a significant contributing element to the annual ecosystem respiration. We also estimate the cumulative ecosystem respiration during the full period of weak thermal stability and demonstrate that it is similar to the periodic total

loss of catotelm porewater CO₂ store (Line 297-302, Figure 6a and c). We consider that these new calculations provide information on 1. The possible average annual porewater CO₂ emission rate, 2. A reasonable range of increase in porewater CO₂ emissions, 3. Indication that porewater CO₂ emission, and their possible change over time, likely contribute to the measured ecosystem respiration at the site. We also no longer state that the GPP and ER partitioning method is flawed.

For the consideration of the possible increase in hydrological CO₂ export, we now present the full time-series of stream CO₂ export (Figure 6b), and estimate the cumulative CO₂ export during the periods of weak thermal stability. These results still confirm that the magnitude of the stream CO₂ export per catchment area is about one order of magnitude lower than the loss of porewater CO₂ in autumn. We nonetheless provide additional references to state that hydrological CO₂ export may be greater at this location than over the rest of the catchment area, owing to the presence of a preferential flow layer at this site (2 to 2.5m deep) and the site's proximity to the stream outlet (i.e. flow convergence area) (Line 317-319).

The reference (Griffiths et al, 2016) is now presented in the discussion (Line 240-241).

This revised version of the manuscript also includes a paragraph stating the possible environmental condition at this site that could make it more prone to turbulence in peat porewater (Line 206-219). We formulate several recommendations for more detailed studies throughout the discussion e.g., 1. the possible interplay between increase in mass flow and turbulent diffusion in autumn (Line 202-204, 249-251), 2. The role of changing porewater CO₂ store in the peatland C budget (Line 302-304, 322-323). We also highlight the latter element of uncertainty in the introduction (Line 65-66). We are currently working on several follow up studies and have instrumented a new peat depth profile near the eddy-covariance tower at Degerö Stormyr and at another nearby and newly-restored peatland.

Specific comments:

Line 36 – reference needed for 'hydro-physically inactive'

Statement removed

Line 41 – Does 'CO₂' refer to concentration or stock? Spelling: 'instantaneous'

Statement revised. Spelling corrected

Line 60 – 'manual characterization with a low time resolution' – rephrase for clarity, e.g., periodic manual measurements?

Sentence corrected (Line 54)

Line 111 (and following) – 'overlying'

Corrected

Lines 120-121 – Is the second value for the 1-2 m layer (not 1.5 m)?

Statement clarified

Lines 124-126 - Near-surface porewater CO₂ concentrations will also be affected by rhizosphere processes, which change seasonally.

Rhizosphere processes are now mentioned (Line 73)

Line 195 / Figure 5b – Given the relationship stated in the equation, it would make more sense for K_z and the trend lines to be plotted against 1/N₂ rather than N₂.

We indicate the direction of weak and strong thermal stability on the axis itself (Figure 4b)

Line 206 – Spelling: example

corrected

Line 227 – Spelling: simultaneous

corrected

Line 285 – Spelling: temporarily

corrected

Line 328 – Spelling: simultaneous

corrected

Line 426 – Spelling: equipped

corrected

Line 431 – Spelling: bungs

corrected

Lines 467-468 – ‘in order to account for dilution...’

corrected

Line 476 – ‘volumetric unit area’ – unclear what this means, as the dimensions are $[\text{mass}] [\text{volume}]^{-1} [\text{time}]^{-1}$

Calculations removed as explained in main comment above

Line 507 – Does this mean 15 days in total?

Calculations clarified (Line 484)

Line 537 – Spelling: dynamics

corrected

Reviewer #3 (Remarks to the Author):

Review of Campeau et al. 'Autumn destabilization of deep porewater CO₂ storage in a northern peatland driven by turbulent diffusion' submitted to Nature Communications.

Overview

This paper is a thought-provoking account of a high-quality dataset from a northern peatland that suggests that CO₂ dissolved in deep (> 0.5-0.75 m depth) porewater can be lost from peatlands during the autumn. Previously, this store of CO₂ has been assumed to be largely static or having a slow turnover. In a mostly well-written paper, the authors suggest that their dissolved CO₂ data can be explained by the process of turbulent diffusion, which can lead to rates of dissolved CO₂ transport that are orders of magnitude greater than molecular transport. In a careful analysis, the authors also consider the importance of the posited process compared to other processes in the carbon cycle in peatlands. Overall, I enjoyed reading the paper and I believe there is information in it that is very worthy of publication in Nature Communications. However, I struggled to understand parts of the manuscript and I believe the authors could do a better job of explaining their proposed mechanism.

Below, I explain how I think the paper could be improved. I have also marked up the pdfs of the manuscript and the SI, and the editor and authors are referred to these for other comments and concerns that I recommend are addressed.

More than one type of turbulent diffusion?

Groundwater scientists usually consider there to be two ways by which solute mixing may occur: molecular diffusion and mechanical dispersion, the latter being analogous to diffusion but occurring in laminar flows due to the 'splitting' of flow through a porous medium as flowing water 'samples' the wide range of pore sizes. Turbulence can sometimes occur in porous media in, for example, biogenic macropores such as wormholes, larger voids between soil peds, and joints in bedrock. Turbulent transfer of a range of properties is also known to occur in surface waters, and the transfer has been found to follow mathematically the diffusion equation; hence, the use of the term 'turbulent diffusion'. The latter occurs when there is rapid mass transfer of water. The authors, however, seem to be suggesting a type of turbulent transfer that does not involve mass flow of water. From the text and Figure 6 they propose that pore-scale turbulent eddies form that help transfer dissolved CO₂ along a concentration gradient. They also note that this turbulence causes a transfer of CO₂ that is orders of magnitude greater than would occur via molecular random walks.

In principle, I find this suggestion plausible, but what I find confusing is the authors' explanation, or lack of explanation, of the mechanisms involved. In short: what causes the turbulence? The authors are clear in suggesting mass flow is not responsible (e.g., on lines 226-7), and they warn against treating the pore-water domain in the same way as lake water where turbulent diffusion has also been observed. Perhaps it is my ignorance showing here, and perhaps other scientists will clearly understand what the authors mean. However, I suspect many target readers of the paper will be like me and not know how turbulent diffusion can occur without mass flow. I guess part of my problem here is that I struggle to see how turbulent mixing at the scale posited by the authors can occur in the pores found in the deeper peat at their site. They suggest that turbulent eddies in the peat pores might form at scales of mm to cm but I don't see how that is possible because most pores in the deep peat will be of mm scale or much less (based on the hydraulic conductivity data presented in the SI). Notably, on lines 227-235, the authors themselves seem somewhat uncertain about how turbulent diffusion can occur in peat in the absence of mass flow.

We thank the reviewer for identifying those aspects which needed further clarification in the manuscript. Indeed, a high level of clarity on the mechanism of turbulent diffusion is essential in this paper. We hope that our changes have made our explanations clearer:

1. Source of turbulence

Turbulence consists of small-scale random fluid motion (eddies) propagating in the peat porewater (described as K_z by Osborn et al, 1980), which is different to directional flow of water through the peat pores (determined by K_{sat} , Figure S2). Vertical gas diffusion in water occurs at the molecular level (slow and constant) when the water is completely still and contains no kinetic energy (generally valid in groundwater). However, the high porosity of the peat porewater, its exposure to wind shear on the peatland surface and flow through the peat pores provide a source of kinetic energy in the peat porewater. Hence, the porewater is not still, and contains a very small level of kinetic energy (propagating as turbulence), which increases vertical diffusion relative to the molecular rate (i.e., turbulent diffusion, variable in time and space). Kinetic energy input provides the source of turbulence, and changes in porewater temperature stratification determines the degree of suppression or propagation of this kinetic energy in the form of turbulence (i.e., small scale eddies) in the catotelm porewater. Weak thermal stratification (weak stability) allows turbulence to propagate, while a strong thermal stratification

(strong stability) suppresses turbulence. This shift in turbulence propagation can occur even under constant and low kinetic energy input.

We improved the discussion (Line 127-219) and method (Line 453-490) sections and the schematic (Figure 5) to further clarify the mechanism of turbulent diffusion. We provide further clarification of the model of turbulent diffusion, described by Osborn (equation 1). We explain the interplay between kinetic energy and changes in porewater thermal stability (equation 1) and discuss the estimated levels in kinetic energy in the porewater (Line 150-158). Figure 4b now displays the estimated kinetic energy levels (oblique lines) in different porewater depths. This reveals that kinetic energy in the peat porewater is very low (on average one order of magnitude lower than in small and sheltered lakes Line 154-156), but also considerably higher in the surficial porewater than deeper peat horizons. We also state the possible source of this kinetic energy (Line 157-158) (i.e., wind shear near the peat surface and lateral flow through the peat pores) and also point to the possible role of ebullition as another source of kinetic energy in the catotelm porewater (Line 265-266, see next comment).

We did not intend to suggest that there was more than one type of turbulent diffusion, but rather that diffusion can occur at other rates than strictly molecular (i.e. turbulent). We emphasize the differences between molecular and turbulent diffusion (e.g., turbulent diffusion is defined as vertical diffusive transport that exceeds the molecular rate ($10^{-7} \text{ m}^2 \text{ s}^{-1}$ for heat transfer) (Line 475-476). Unlike molecular diffusion, which is constant and slow, the rate turbulent diffusion can vary widely in time (Line 130-132). We also highlight that the relationship between our estimates of K_{app} and N^2 , to clarify the link between turbulent diffusion and the weakening thermal stratification (Figure 4b, Line 160-170).

The schematic (Figure 5) now illustrates the models of all three diffusion rates a) molecular b) turbulent under strong thermal stability, c) turbulent under weak thermal stability. This makes the differences between molecular and turbulent diffusion clearer. The turbulence in porewater (drawn in black arrows) is now better illustrated in the form of “random fluid motion”. The depth profile now indicates the k_{sat} profile (grey arrows), interpreted from the bulk density measurements at the site. The turbulent diffusion is indeed separate from directional flow of water (i.e., mechanical dispersion (or mass flow) via convection or advection) (Figure 5).

2. Scale of eddies

In the previous version of this manuscript, we mistakenly stated that the scale of the eddies could be of mm to cm scale, while these are in fact the sizes of large eddies measured in lakes. The estimated K_{app} , and kinetic energy in the peat porewater (equation 1) are on average one order of magnitude lower than in small sheltered lakes (Line 154-155). We have rectified this statement, now stating the scale of the eddies is lower than mm scale, which is more consistent with the pore size in the catotelm and the literature (Line 187-189).

3. Properties of the studied site

We added a paragraph in the discussion stating the possible properties of our study site that could make this location more prone to turbulence. This includes 1. Relatively high peat porosity, 2. Presence of a deep preferential flow path and in close proximity to the stream (flow convergence area), which might increase specific CO_2 export. 3. Peatland exposed to strong wind since it is located at a topographic high point (Line 206-219).

Ebullition and measurement of dissolved CO_2

I wonder too whether bubble formation and transport within the peat might also offer a part explanation of the authors' findings. Biogenic gas bubbles tend to accumulate in peat during the growing season and reach a peak unit volume by late summer and early autumn. These bubbles contain CO_2 as well as CH_4 and their movement upwards through the peat profile may also facilitate CO_2 transfer. Their movement through, and loss from, the peat profile will also cause mixing of pore water that might give rise to turbulent diffusive transfer of dissolved CO_2 . I should add that sudden episode of gas bubble loss from peat are not always easily detected or measured by eddy correlation. The flux tower used by the authors was also somewhat distant from the monitored peat profile so may have missed ebullition events near the sensors.

We agree with the reviewer that ebullition could play a role in the porewater CO_2 dynamics. Yet due to the relatively high solubility of CO_2 , most of the CO_2 pool is in dissolved rather than free-phase. Diffusion should therefore be the main transport pathway for CO_2 , despite a share of the gases possibly transported in bubbles. We now discuss the potential role of ebullition in the porewater CO_2 dynamics observed in our data (Line 261-268). We also indicate that the ebullition could provide an additional source of kinetic energy and turbulence in the catotelm porewater (Line 266-268). However, we still ascribe the increase in turbulent diffusion with the weakening of porewater thermal stability as the main process responsible for the sudden losses in catotelm porewater CO_2 store in autumn (Line 268-270).

I have a slight concern about the measurement of dissolved CO₂. What happens to the CO₂ sensors if bubbles are in contact with the sensor membrane? Is it possible that the sensors measure a combination of dissolved and free-phase CO₂?

The sensor could measure a combination of dissolved and free-phase CO₂ (e.g., if bubbles were stuck on the membrane). But as stated previously, the majority of the CO₂ pool should be in dissolved phase.

Also, if bungs were fitted to the tops of the sampling tubes, how did flow of water into and out of the tubes occur? Such tubes need to be vented to allow water ingress and egress.

The tubes have thin slits screening for specific depths (Line 406-408). Flow in and out of the tubes should be proportional to peat hydraulic conductivity at those specific screened depths.

The tubes were “vented” (bung removed) in Spring 2015 and 2016, while the membrane of the sensor was replaced (now stated Line 413-415). We noticed large decreases in CO₂ concentration following those periods where the tubes were left open for several hours/days. The tubes probably acted as chimney for porewater gas evasion to the atmosphere. The CO₂ concentration returned to their initial levels several days/weeks after the tubes closed, which resulted in a large gap in a time-series, specifically during spring. We subsequently refrained from leaving the tubes open for extended periods of time. In the following year (2017), the tubes remained closed in order not to affect conditions during spring thaw.

Tubes were also opened for a few minutes/hours periods when sensors had to be retrieved for winter (every November, near the surface only) or when sensor damaged forces us to redistribute the sensors at different depths follow sensor malfunction. We did not notice large changes in CO₂ concentrations following those shorter manipulations.

Recommendation

Overall, I recommend the paper is revised and reviewed again. I think the authors need to explain their proposed mechanism more clearly, so that wider readership can understand it. Currently, I have some doubts about what the authors are proposing, and I'd like to be more convinced that their mechanism is sound.

We thank the reviewer for his constructive comments and hope that the revised version of the manuscript will adequately answer his concerns.

Please note that I operate a policy of 'open reviewing'. My comments above, and on the marked-up manuscript and SI, are for the editor and the authors and I would like my identity to be revealed to the authors.

Comment by lines:

Line 33. Why latent?

Statement removed

Line 35 Avoid using 'this' in this way (i.e., say what 'this' refers to). 'This lack of oxygen ...'.

Statement corrected

Line 37: Unclear in two ways. What processes are implied by 'hydro-physically'? 'lower depths' suggests shallower (nearer the surface) peat. Don't you mean greater depths?. By whom? I'm not sure there is such a general perception.

We removed this statement. Peat hydrology is indeed dynamic. We rather claim that the processes driving the peatland CO₂ cycling (i.e., atmospheric exchange and hydrological export) operate mostly within the acrotelm (Line 32-44).

Line 52: I think there is a misunderstanding here. Horizontal flow can only occur if there is vertical recharge. The latter then becomes the rate limiter. Vertical and horizontal K cannot be separated from each other when calculating flow through an aquifer, which is not the same as uni-directional flow through a permeameter sample of peat (for example).

Statement removed

Line 58: 'low *rates of* hydrological...'

Statement revised

Line 61: Just say 'store' here.

“Storage” changed to “store” here and throughout the text

Line 68: 'recurring'

Corrected here and throughout the text

Line 75: 'recurring'

corrected

Line 87: It would help if this were defined here.

Clearer definition now provided (Line 90-107). We also changed the terminology from thermal stability and instability to strong/weak thermal stability.

Line 89: Okay, so what mechanism(s) is(are) involved?

We now provide more clarification of the mechanism of turbulent diffusion and the interplay with thermal stability here and throughout the text (see response to first comment)

Line 95: Rather vague wording.

Exact dates with length of the period are now specified (Line 84)

Line 97: Why is it a thermal instability? What is meant by 'instability' here?

We explain the link between strong/weak thermal stratification and thermal stability (Line 91-94). We specify thermal stability because these calculations are based on temperature measurements, but stability (buoyancy) can also be affected by solute concentration. We also refer to strong/weak thermal stability instead of thermal stability and instability.

Line 116: Just the CO₂ or the porewater too?

We hope our improvement to the text have helped clarify this aspect. Since we refer to diffusion and not large-scale directional fluid motion (only small-scale random fluid motion (turbulence))(Line 138-139), it is just the CO₂ that diffuses upwards due to its gradient (Line 128-140).

Line 127: -7d?

now clarified 7 days moving window (Caption Figure 2)

Line 134: But if hydraulic conductivity is low, density-driven flow of water will also be low.

We do not invoke large-scale density-driven flow of water (Line 198-201, Figure S5), only small-scale random fluid motion in the peat porewater (Line 138-139).

Line 137: How so? I don't follow.

We clarify the possible influence of ice-cover on porewater turbulence (Line 177-179)

Line 158: Minus signs missing here.

All depths are now referred to without minus sign

Line 161: There is a lot of information in each plot. Would it be better to select just a few key temperature profiles and use these to illustrate the point being made?

The plots (Figure 3) now show key examples, which we agree, makes the figure much clearer.

Line 186: I agree. However, it would be good to explain a little more about turbulent diffusion here - please see my report. The paper could be made more accessible to those, like me, who were unaware of the proposed mechanism.

We added more explanations in the text as well as improved the schematic Figure 5

Line 202: Four orders of magnitude?

corrected

Line 203: 'Such changes imply...'

corrected

Line 206: 'example'

corrected

Line 225: What implies? Unclear. Avoid using the 'unspecified' 'this'.

Statement revised accordingly

Line 227-235: Why not? I don't follow the argument here. I find the argument here difficult to follow and understand.

We reworked this section of the text to explain that the increase in vertical diffusion in autumn may not be accompanied by an increase in convection, as referred to in lakes with turbulent mixing (Line 186-204).

Line 241: But is there not threshold too at about 4 deg. C below which respiration shuts down?

We now provide a reference stating that respiration can even peak at -5°C (Line 229)

Line 244-245: Unless they cause ebullition. Also, in what way are the deeper waters confined? If the peatland is a water-table aquifer then it is unconfined.

Statement revised.

Figure 6: This is a nicely-designed figure, but I think it is misleading to suggest lateral export at 2 m depth is the same as that near the surface. Perhaps I have misunderstood something here. Also, shouldn't the CO₂ emission be larger in the right-hand figure?

Figure edited (see response to first comment)

Line 289: The surface porewater is not hydrologically conductive; the peat is. Sentence needs rewording.

Statement corrected

Line 326: Actually, it could occur from a range of depths. There is a lot of work that shows CH₄ production at shallow depths too; e.g., just below the water table where there is abundant substrate and also sufficient anoxia.

Calculations in this section have been removed (See response to Reviewer 2)

Line 330: Okay, but what about release of methane-containing bubbles?

We discuss the potential role of ebullition (Line 261-270)

Line 333: Profile of what?

Corrected

Line 397: I agree this is very low. It is not typical of peatlands more widely. Was the study site therefore quite unusual?

We now present a section discussion the particularities of our studied location in connection with the mechanism of turbulent diffusion (Line 206-219).

Line 407: Name / proper noun? In which case start the words with capitals?

Corrected

Line 431: 'bung's', But the bungs would also have affected water flow into and out of the tubes, so how can you be sure that the water in the tubes was in equilibrium with the pore water?

We assume that the speed of water inflow and outflow through the tubes is proportional to the hydraulic conductivity at the porewater layer screened by the tube. We measured considerable seasonality in porewater solute concentration and isotopic ratio (Campeau et al 2017 and 2018), and water temperature (Figure 1b), which indicates continuous water recharge in the tubes. We agree with the reviewer that there is always a level of uncertainty when undertaking measurement in groundwater tubes. However, deploying the sensors (with a fragile membrane) would not have been possible without those groundwater tubes. We recently launched a follow up study where we have instrumented another peat depth profile in close proximity with the eddy-covariance tower. For this new instrumentation, we invested in new and more sturdy sensors (eosense GP), which we could deploy directly in the peat, without requiring the installation of groundwater tubes.

Line 467: 'to account for'

Corrected

Line 476: m-2?

Corrected

Line 531: Okay, but was there any evidence of enhanced ebullition during the autumn period?

Not to our knowledge, but a more detailed examination of CH₄ ebullition at the site is planned for the future.

Figure S2: The figure caption says m below ground surface. Also, there is the potential for confusion with the use of negatives. A negative depth (below ground) means something is above the surface. I suggest being clearer in the axis label (something like 'water table position relative to the ground surface' would be okay). Also, why does

the water table rise above the surface during the ice/snow covered period. Is the peat at that time actually a confined aquifer when the concept of water table does not apply?

Depth reporting are adjusted here and throughout the text.

The positive water table measurements in winter are caused by the accumulated snow and ice cover putting additional pressure on the pressure transducer. These measurements are highly uncertain, so they have been removed. These data are now presented in Figure 1d.

Figure S3: What measurement method was used?

The methods are detailed in [Nijp *et al.*, 2019]. We nonetheless specify that they were inferred based on piezometer measurements of water table depth.

Andy Baird,
Chair of Wetland Science
University of Leeds, UK;
13th April 2021.

Additional Changes

- Removed monthly measurements of porewater DOC concentration from Figure S5, since they did not convey more information than the water stable isotope measurements. These data are already presented in (Campeau et al, 2017 and 2018).
- Changes the heatmap of porewater temperature for the time-series, to show more clearly periods of temperature equilibration, and for comparative use with other studies in the future
- Main text includes the water table position time-series (Figure 1d) since this is usually an important variable in peatland studies.

REVIEWER COMMENTS

Reviewer #1 (Remarks to the Author):

The authors have done a good job of responding to all of the points raised by the reviewers. I do not have any additional significant concerns. I would suggest some minor revisions to the figures to make them more legible. For example Figure 1 I would recommend that the text of the figure legends be a larger font size. The same for the text on the panels in Figure 3; they were difficult to read at 100%. For figure 3, I might also suggest making the two series a different symbol shape as well as color to further distinguish them. Finally there appears to be a typo on line 530 should be "atmosphere".

Reviewer #2 (Remarks to the Author):

General comments: Turbulent diffusion is a new idea for peatland scientists, and the evidence and explanation presented in this manuscript clearly demonstrate its potential implications for peatland carbon cycling. The revisions improve the manuscript by clarifying the proposed mechanism of turbulent diffusion and by providing a more cautious appraisal of the implications for CO₂ emission and water-borne export.

Specific comments:

Lines 43-44 – ‘contribution to the peatland CO₂ cycling’ is a bit vague, and I’m not sure what it would mean for this contribution to be ‘low’

Lines 52-53 – not a sentence (needs a verb)

Lines 54-55 – This paragraph needs to end on a clearer statement of the message, e.g., assumption of slow dynamics and sampling at coarse temporal resolution (state timescales?) may miss seasonal dynamics driven by process(es) other than diffusive transport?

Lines 153-154 – check grammar

Line 176 – leads to?

Lines 201-202 – stable isotope ratio of what? (13C)O₂?

Lines 247-248 – ‘water stable isotope ratio’ of what? D₂O? H₂(18O)?

Line 280 – temporally, rather than temporarily?

Line 291 – delete ‘on’

Lines 295-296 – may hold a significant share...

Line 300 – corresponding to average concentrations...?

Line 423 – bungs?

Line 425 – water level measurements

Line 455 – summing

Line 457 – volume

Line 458 – included

Line 520 – ebullitive – not captured?

Line 523 – the

Line 533 – ‘plant community composition’ = vegetation

Reviewer #3 (Remarks to the Author):

Review of Revision 1 of Campeau et al. ‘Autumn destabilization of deep porewater CO₂ storage in a northern peatland driven by turbulent diffusion’ submitted to Nature Communications.

Overview

I reviewed the original submission of this paper and have been asked to comment on whether the revision takes account of the concerns raised in my original report. I also read the reports of the other two reviewers, and the authors’ explanations of how they responded to the first round of reviews. Overall, I think this revision is a considerable improvement on the original. In particular, it is much clearer in its explanation

of turbulent diffusion and the mechanisms involved. However, I think some further changes will make it more convincing. Upon re-reading the work, I have also identified a possible problem with the data set. My more substantive concerns are articulated below. I have also added comments to a pdf of the paper to which the editor and authors are referred.

CO₂ production in autumn and winter?

I apologise for not discussing this issue in my first review, but, looking at the data again, I am struck by the rapid increases in porewater [CO₂] in the late autumn and early winter. The authors interpret the rapid late-summer and early-autumn declines in porewater [CO₂] as a substantial loss of CO₂ from their monitored peat profile, but where does the CO₂ come from for the rise that follows? What plausible mechanisms can be invoked? CO₂ production will occur slowly in the catotelm because of anoxia and will also be slow in the acrotelm as temperatures fall. This apparent discrepancy needs to be resolved if we are to believe the data. The authors did not measure 3-D patterns of porewater [CO₂], and it is possible that CO₂ is moving laterally in a complex way through the peat profile, and such movement may help explain the data. In other words, the 1-D patterns observed by the authors may not apply laterally across the peatland.

Piezometer tubes

In my original review I noted: "Also, if bungs were fitted to the tops of the sampling tubes, how did flow of water into and out of the tubes occur? Such tubes need to be vented to allow water ingress and egress." In their rebuttal, the authors note "The tubes have thin slits screening for specific depths (Line 406-408). Flow in and out of the tubes should be proportional to peat hydraulic conductivity at those specific screened depths." In the revised paper they also note: "The top of the wells was sealed with thick rubber bongs [sic] to prevent atmospheric gas exchange."

If a well is screened below the lowest position of the water table and is sealed at the top of the lining tube, it will not function properly as a piezometer, regardless of the hydraulic conductivity and hydraulic gradients in the soil. Air trapped in the well tube will act to stop water ingress and egress. If, for example, porewater pressures rise outside the screened section, water will flow into the well to equilibrate to the new pressure. However, as water begins to flow into the piezometer it will start to compress the air, which becomes pressurised, thus preventing more water entering the tube. A similar argument applies when porewater pressures in the soil around the piezometer fall. This problem is well known among hydrogeologists, but it is unclear whether it produced artefacts or errors in the authors' porewater [CO₂] data. I think we can be confident that water flow into and out of the tubes was inhibited and, therefore, that the water around the CO₂ sensors will have been 'old' water. However, I suspect the distance between the sensor membrane and peat outside the piezometer (of the order of 8 mm allowing for tube wall thickness) was probably short enough to mean dissolved CO₂ by the membrane was in quasi equilibrium with porewater CO₂ in the peat (equilibration occurring via molecular and turbulent diffusion).

Thermally-driven convective flow?

On line 480, the authors correctly note that the hydraulic conductivity of the catotelm is almost certainly too low for thermally-driven convective flow of porewater to occur (see, for example, <https://www.pnas.org/content/100/25/14937.short>). However, in saying this the authors appear to contradict what they say on lines 207-208. I recommend they rewrite what is on those lines so that it is consistent with what they say later in the paper.

Soil respiration at 3-5°C

On lines 231-235, the authors note: "The temperature dependency of peat soil respiration is mostly linear and thus inconsistent with a rapid shutdown at 3-5 °C 39, 40, 41, which would be necessary to explain the phenomenon in autumn revealed by our data. One study even reported peak CO₂ production in peat porewater around 5°C 42, which would be opposite to the losses in catotelm porewater CO₂ store observed each autumn in this peatland." I'm still not convinced by what is written here – it reads a little like special pleading. There is a huge body of knowledge that shows that ecosystem respiration (ER) in peats is low or shuts down at ~4° C – see, for example, the very large flux chamber literature (the authors' own ER data also show this). I know ER includes aerobic decay in the acrotelm and plant respiration, but it also includes anaerobic microbial respiration in the catotelm. I think the discussion here could be more nuanced/balanced.

Units

In several places units don't seem to be given correctly or are not given at all. For example, at times I was confused about whether the diffusive flux or the diffusion coefficient was meant. Elsewhere, units for kinetic energy dissipation are seemingly used for kinetic energy.

I continue to operate a policy of 'open reviewing'. My comments above, and on the marked-up manuscript, are for the editor and the authors and I would like my identity to be revealed to the authors.

Andy Baird,
Chair of Wetland Science
University of Leeds, UK;
5th August 2021.

Reviewer #1 (Remarks to the Author):

The authors have done a good job of responding to all of the points raised by the reviewers. I do not have any additional significant concerns. I would suggest some minor revisions to the figures to make them more legible.

We thank the reviewer and are happy that he/she is satisfied with our revision of the manuscript

For example Figure 1 I would recommend that the text of the figure legends be a larger font size. The same for the text on the panels in Figure 3; they were difficult to read at 100%.

Font size is increased in both figures

For figure 3, I might also suggest making the two series a different symbol shape as well as color to further distinguish them.

Symbol for temperature data is now different from the CO₂ concentration data

Finally there appears to be a typo on line 530 should be "atmosphere".

Corrected

Reviewer #2 (Remarks to the Author):

General comments: Turbulent diffusion is a new idea for peatland scientists, and the evidence and explanation
presented in this manuscript clearly demonstrate its potential implications for peatland carbon cycling. The
revisions improve the manuscript by clarifying the proposed mechanism of turbulent diffusion and by providing a
more cautious appraisal of the implications for CO₂ emission and water-borne export.

We thank the reviewer and are happy that he/she is satisfied with our revision of the manuscript

Specific comments:

Lines 43-44 – ‘contribution to the peatland CO₂ cycling’ is a bit vague, and I’m not sure what it would mean for
this contribution to be ‘low’

Sentence now rephrased (Line42-43): *“As long as CO₂ remains confined in the catotelm, its role in the*
*peatland CO₂ cycling is negligible”*

Lines 52-53 – not a sentence (needs a verb)

Corrected (Line 51)

Lines 54-55 – This paragraph needs to end on a clearer statement of the message, e.g., assumption of slow
dynamics and sampling at coarse temporal resolution (state timescales?) may miss seasonal dynamics driven by
process(es) other than diffusive transport?

New sentence added at the end of the paragraph to more explicitly define the knowledge gap (Line 54-
56): *“The assumption of slow dynamics in catotelm porewater gas store has not been explicitly tested, which may*
*result in overlooking key process controlling temporal dynamics in peatland CO₂ cycling. “*

Lines 153-154 – check grammar

Corrected

Line 176 – leads to?

Corrected

Lines 201-202 – stable isotope ratio of what? (13C)O₂?

Corrected

Lines 247-248 – ‘water stable isotope ratio’ of what? D₂O? H₂(18O)?

Corrected

Line 280 – temporally, rather than temporarily?

Corrected (here and throughout the text)

Line 291 – delete ‘on’

Corrected

Lines 295-296 – may hold a significant share...

Corrected

Line 300 – corresponding to average concentrations...?

Corrected

Line 423 – bungs?

Corrected

Line 425 – water level measurements

Corrected

Line 455 – summing

Rephrased (Line 489-490): “*The total porewater CO₂ store for the top two meters of the peat depth profile was*

*estimated by deriving the sum of the porewater CO₂ store for three individual porewater layers (i.e. [0 to 0.5 m],*

*[0.5 to 1 m], [1 to 2 m]).*”

Line 457 – volume

Corrected

Line 458 – included

Corrected

Line 520 – ebullitive – not captured?

Replaced by measured

Line 523 – the

Corrected

Line 533 – ‘plant community composition’ = vegetation

Corrected

Reviewer #3 (Remarks to the Author):

Review of Revision 1 of Campeau et al. 'Autumn destabilization of deep porewater CO₂ storage in a northern
peatland driven by turbulent diffusion' submitted to Nature Communications.

Overview

I reviewed the original submission of this paper and have been asked to comment on whether the revision takes
account of the concerns raised in my original report. I also read the reports of the other two reviewers, and the
authors' explanations of how they responded to the first round of reviews. Overall, I think this revision is a
considerable improvement on the original. In particular, it is much clearer in its explanation of turbulent diffusion
and the mechanisms involved. However, I think some further changes will make it more convincing. Upon re-
reading the work, I have also identified a possible problem with the data set. My more substantive concerns are
articulated below. I have also added comments to a pdf of the paper to which the editor and authors are
referred.

We thank the reviewer for his constructive feedback. A detailed response to his concerns follow:

1. CO₂ production in autumn and winter?

I apologise for not discussing this issue in my first review, but, looking at the data again, I am struck by the rapid
increases in porewater [CO₂] in the late autumn and early winter. The authors interpret the rapid late-summer
and early-autumn declines in porewater [CO₂] as a substantial loss of CO₂ from their monitored peat profile, but
where does the CO₂ come from for the rise that follows? What plausible mechanisms can be invoked? CO₂
production will occur slowly in the catotelm because of anoxia and will also be slow in the acrotelm as
temperatures fall. This apparent discrepancy needs to be resolved if we are to believe the data. The authors did
not measure 3-D patterns of porewater [CO₂], and it is possible that CO₂ is moving laterally in a complex way
through the peat profile, and such movement may help explain the data. In other words, the 1-D patterns
observed by the authors may not apply laterally across the peatland.

We agree with the reviewer that the rate of recovery in early winter needed more formal analysis (now
presented on Lines 197-211). We now provide further information on the recovery of porewater CO₂ in
early winter (only measured at 0.75 m and 1.5 m depth). Figure 1 in the response letter (now Figure S1 in
supplementary files) indicate that the speed of recovery at both depths was generally slow and steady (on
average 0.05 (SD±0.12) and 0.04 (SD±0.07) mg C g⁻¹ d⁻¹ at 0.75 m and 1.5 m, respectively, throughout
the 60 days following the last drop in porewater CO₂ concentration in autumn across all four years). This
rate of recovery is reasonable based on laboratory incubation of peat samples in cold temperatures (Segura
et al., 2019, Waddington et al., 2001 and Kolton et al., 2019). Average rates of recovery for this 60 days
period are presented in detail for each year in Table S1 in the supplementary files.

There were, however, two years where the recovery in porewater CO₂ at 0.75 m was faster (i.e. 2014 and
2017, Figure 1a of response file & Figure S1a of supplementary file). The rate of recovery was above 0.2
132 mg C g⁻¹ d⁻¹ for 5 days in 2014 and 2 days in 2017, with a maximum of 1.2 mg C g⁻¹ d⁻¹ (Figure 1 in
response letter). We consider that those high rates of CO₂ increase are likely the result of additional
transport processes supplementing biogenic production. In those same two years (unlike the other years)
the porewater CO₂ concentration at 1.5 m depth decreased sharply during autumn, indicating that the
destabilisation of porewater CO₂ store reached deeper into the peat profile in those years. A likely
explanation is that the more severe destabilisation of the catotelm porewater CO₂ store on those years was

caused by higher turbulence or a prolonged period of thermal instability, which could allow CO₂ from deeper peat horizons to diffuse upward, or laterally from other areas of the peatland, and assist in the recovery of porewater CO₂ at 0.75 m depth.

Figure 1: Timeseries of a,b) porewater CO₂ concentration (mgC L⁻¹) and c,d) the corresponding daily changes in porewater CO₂ concentration (mg C g⁻¹ d⁻¹) between September and December. Each line with circles represents a different year. (Added as Figure S1 in the supplementary file in the revised version of the manuscript)

Worth noticing is also that the annual temperature cycle in the deep porewater (0.75 m and 1.5 m) is considerably delayed compared with the surface porewater temperature (Figure 2 of response letter). At 0.75 m and 1.5 m, the highest porewater temperatures are reached in the autumn (Aug to Oct) and remain above the annual average until December. This time lag in temperature may contribute to sustaining relatively high local CO₂ production rates during the late autumn - early winter.

*Figure 2: timeseries of the porewater temperature (°C) at a) 0.75m and b) 1.5m between January and December with*
*each line representing a different year. The black horizontal line indicates the annual average porewater temperature.*

We also added a few sentences at the beginning of the results and discussion section (line 73–77) to
remind the readers that the concentration timeseries reflect the net balance between CO₂ input (transport
and production) and outputs (mostly transport). We consider that it was useful to set the basis for further
discussion on CO₂ loss and recovery.

**2. Piezometer tubes**

In my original review I noted: “Also, if bungs were fitted to the tops of the sampling tubes, how did flow of water
into and out of the tubes occur? Such tubes need to be vented to allow water ingress and egress.” In their
rebuttal, the authors note “The tubes have thin slits screening for specific depths (Line 406-408). Flow in and out
of the tubes should be proportional to peat hydraulic conductivity at those specific screened depths.” In the
revised paper they also note: “The top of the wells was sealed with thick rubber bongs [sic] to prevent
atmospheric gas exchange.”

If a well is screened below the lowest position of the water table and is sealed at the top of the lining tube, it will
not function properly as a piezometer, regardless of the hydraulic conductivity and hydraulic gradients in the soil.
Air trapped in the well tube will act to stop water ingress and egress. If, for example, porewater pressures rise
outside the screened section, water will flow into the well to equilibrate to the new pressure. However, as water
begins to flow into the piezometer it will start to compress the air, which becomes pressurised, thus preventing
more water entering the tube. A similar argument applies when porewater pressures in the soil around the
piezometer fall. This problem is well known among hydrogeologists, but it is unclear whether it produced
artefacts or errors in the authors’ porewater [CO₂] data. I think we can be confident that water flow into and out
of the tubes was inhibited and, therefore, that the water around the CO₂ sensors will have
been ‘old’ water. However, I suspect the distance between the sensor membrane and peat outside the
piezometer (of the order of 8 mm allowing for tube wall thickness) was probably short enough to mean dissolved
CO₂ by the membrane was in quasi equilibrium with porewater CO₂ in the peat (equilibration occurring via
molecular and turbulent diffusion).

This is an important point, and we agree with the reviewer that we could have described it better.

However, we argue that this impact on our data and analysis was minor. The major reason is,
as also pointed out by the reviewer, that the very short distance (6.5 mm) between the sensor membrane
and the external porewater allowed for continuous and rapid diffusive gas exchange between the
surrounding peat porewater and water inside the tube, despite a possible effect on water ingress and
egress. Our general findings of the importance of turbulent diffusion in peat porewater are supported by

porewater gas concentration and temperature measurements, which would equilibrate by diffusion
between the outside and inside of the tubes, regardless of possible reduction in water transport.

Furthermore, the amplitude in groundwater table fluctuation across the year at our study site is very low.
The lowest water table depth is 19 cm relative to ground surface over the four years of observation. Under
0.2 meter of fluctuation in groundwater table will not give the type of pressure build up in the top-sealed
tubes that can be seen in some hydrogeological settings where meters of variation in groundwater table
can take place. Therefore, we consider that variations in air pressure in the tubes will have a minor
influence on the water exchange.

We nonetheless explicitly mention this possible artefact of the groundwater tubes on water ingress and
egress, and further highlight that the importance of the closeness between the sensor and the outside of
the tube in generating reliable measurements of porewater gases and temperature equilibrating by
diffusion. (Line 457-463)

3. Thermally-driven convective flow?

On line 480, the authors correctly note that the hydraulic conductivity of the catotelm is almost certainly too low
for thermally-driven convective flow of porewater to occur (see, for
example, <https://www.pnas.org/content/100/25/14937.short>). However, in saying this the authors appear to
contradict what they say on lines 207-208. I recommend they rewrite what is on those lines so that it is consistent
with what they say later in the paper.

We thank the reviewer for pointing out the apparent contradiction, and for providing this reference. We
have now added this information to the manuscript and removed sentences suggesting uncertainty in the
possibility of convective mixing (Line 230-231). With these revisions we trust that our arguments are
now internally consistent.

4. Soil respiration at 3-5°C

On lines 231-235, the authors note: “The temperature dependency of peat soil respiration is mostly linear and
thus inconsistent with a rapid shutdown at 3-5 °C 39, 40, 41, which would be necessary to explain the
phenomenon in autumn revealed by our data. One study even reported peak CO₂ production in peat porewater
around 5°C 42, which would be opposite to the losses in catotelm porewater CO₂ store observed each autumn in
this peatland.” I’m still not convinced by what is written here – it reads a little like special pleading. There is a
huge body of knowledge that shows that ecosystem respiration (ER) in peats is low or shuts down at ~4° C – see,
for example, the very large flux chamber literature (the authors’ own ER data also show this). I know ER includes
aerobic decay in the acrotelm and plant respiration, but it also includes anaerobic microbial respiration in the
catotelm. I think the discussion here could be more nuanced/balanced.

We agree with the reviewer that this section needed to be better articulated. We have reworked this
section and provided further support for our claim that dropping porewater CO₂ production alone cannot
explain the losses in porewater CO₂ present in our data (Line 254-267). This section includes the
following arguments:

- 1. Temperature response of peat porewater CO₂ production is often linear (Wu et al, 2021, Scanlon
et al., 2000, Bergman et al., 1999), and sometimes inconsistent (e.g. double maxima due to
shifting metabolic pathways) (Kolton et al., 2019).
- 2. The amplitude in porewater temperature fluctuation in autumn is low, with maximum
temperature reached between August and October, and declines slowly until December.
- 3. The ER fluxes in autumn are still considerably high at this site and suggest no sudden shut down
during the periods of rapid loss in catotelm porewater CO₂ store.
- 4. The rate of porewater CO₂ loss in autumn exceeds the highest measured peat decomposition rate
indicating that a complete shutdown of peat decomposition cannot possibly explain the observed
losses in catotelm porewater CO₂ concentration.

As noted by the reviewer, the contribution from catotelm heterotrophic respiration should be smaller in
comparison with the autotrophic (above and belowground) and the acrotelm (mostly aerobic)
heterotrophic respiration. The rapid shutdown in ER around average daily temperature of ~4°C is often
attributed to plant senescence following the first episode of soil frost at night. For these reasons, the

temperature sensitivities for peatland ER is poorly comparable with catotelm CO₂ production dynamics,
which are the focus of this paper.

5. Units

In several places units don't seem to be given correctly or are not given at all. For example, at times I was
confused about whether the diffusive flux or the diffusion coefficient was meant. Elsewhere, units for kinetic
energy dissipation are seemingly used for kinetic energy.

All units have been double checked and revisions made where appropriate (see line by line comments)

I continue to operate a policy of 'open reviewing'. My comments above, and on the marked-up manuscript, are
for the editor and the authors and I would like my identity to be revealed to the authors.

Line by Line Comments:

Line 15: This is a specialist term used and understood only by peatland scientists. Perhaps define it here for more
general readers.

We agree with the reviewer that the term catotelm is not appropriate in the abstract. Due to the limited
word count, it was difficult to fit in the definition of catotelm. Hence, we simply removed the term and
kept our explanation to "deep porewater" which is the term that is also used in the title

Line 16: Do you actually mean 'temporally' (over time)? 'temporarily' means for a short period of time. Perhaps
just delete because 'stable' alone is clear.

"Temporarily" Removed

Line 21: This last part of the sentence doesn't quite make sense. The sentence is about CO₂ transfer, not water
flow. Okay, after reading the rest of the paper, this point is now clear, but it won't be to someone who has only
read the abstract. The abstract should make sense without someone having to read the paper, so I suggest
improving the wording here

Atmospheric emission and hydrological export are replaced by "net losses" for better clarity in the abstract
(Line 21)

Line 35: So do non-vascular plants.

"Vascular" removed

Line 60: This sentence would read better if this phrase was moved to appear after "hourly measurements".

Corrected

Line 61: recurring

Corrected

Line 64: Add what is it caused by?

Added that turbulent diffusion is caused by the presence of small-scale random fluid motion, with
reference to Imboden et al., 1995 (Line 65)

Line 65: The process has always occurred - it's not new - but was hitherto unknown. Change wording as follows:
'reveal a hitherto unknown process that...?'

Corrected (Line 68)

Line 74: rhizospheric

Corrected (Line 80)

Line 87-90: How is this possible? How is the CO₂ produced over the winter? This sentence seems to contradict

the previous one.

See response to comment 1 above. We added a section providing a more detailed interpretation of the

CO₂ recovery during winter (Line 197-211), We also present the detailed time series of porewater CO₂

concentration and daily changes in concentration at 0.75 m and 1.5 m throughout autumn and early-

winter, indicating that most periods of recovery can explained by local biogenic CO₂ production (Figure

S1). Table S1 also details the rate of CO₂ recovery in early-winter at both depths for each individual year.

With respect to the annual temperature variation, maxima occur between August and October at 0.75m

and 1.5m depth. With respect to the kinetic constraints on peat decomposition, the highest annual

production rates possibly occur during autumn.

Line 106: decreases

Replaced by “which brings N² near zero” (Line 115)

Line 133: Are you referring to diffusive flux here or to the value of the diffusion coefficient? -2?

Coefficient now indicated (Line 140)

Line 135: -2?

This line refers to the kinetic energy input which is expressed in m² s⁻³

Line 154: Delete.

Corrected

Line 157: Incorrect units?

Units checked and they were correct: kinetic energy dissipation rate (m² s⁻³)

Line 176: leads to

Corrected

Line 191: Almost certainly?

Replaced Likely by “almost certainly”

Line 191: Probably much lower than mm scale.

Replaced below by “probably much lower”

Line 208: Studies have been done on thermally-generated convection in shallow peats and for convection to

occur the peat has to have a very high hydraulic conductivity. See, e.g.:

<https://www.pnas.org/content/100/25/14937.short>

We thank the reviewer for providing this piece of information. The results of this study are now explicitly

mentioned on Line (225-227).

Line 214: It doesn't cause a higher porosity; it is associated with it.

Replaced "results in" by "is associated with".

Line 221: At the landscape scale? If the site occurs at a flow convergence, it can't be at a high-point locally.

Added "in the landscape"

Line 226: Simplify this sentence and join with the next sentence?

Sentences merged (Line 250-252): "*Other factors, such as changes in biological CO₂ production, water*
*table position, air-water gas exchange velocity and ebullition contribute to the temporal variability in*
*porewater CO₂ concentration.*"

Line 228: Why and/or?

Changed to "and"

Line 230-235: I'm still not convinced by what is written here - it reads a little like special pleading. There is a huge
body of knowledge that shows that ecosystem respiration (ER) in peats is low or shuts down at ~4 deg. C - look at
the very large flux chamber literature. I know ER includes aerobic decay and plant respiration, but it also includes
anaerobic microbial respiration in the catotelm. I think the discussion here could be more balanced.

This statement has been revised (Line 254-267), see response to general comment 4 above

Line 238: Remove comma.

Removed

Line 259:265: Okay, but wind shear is suggested earlier as responsible for adding kinetic energy to the system to
increase turbulent diffusion. Wouldn't you therefore expect a relationship between wind speed and porewater
CO₂? Are you saying the thermal stability of the porewater masks the effect of wind?

We have added further explanations (Line 292-295), including the following statement; "*While wind is*
*most likely an important source of kinetic energy in the peat porewater, its effect on turbulence is greatly*
*modulated by the thermal stability.*"

Line 280: This phrase (to me at least) is not very clear. See my earlier comment about 'temporally' vs
'temporarily'. English is a fickle language!

We thank the reviewer for identifying these mistakes. Temporarily changed for temporally.

Line 304: A flux is a rate, so just say 'fluxes' here?

Corrected

Line 327: ca. - just one stop (abbreviation for Latin circa).

Corrected

Line 330: I don't quite follow the argument here. From the CO₂ export figure you give in the previous paragraph,
it seems losses via streamflow are far too low to account for an appreciable proportion of the loss of CO₂ inferred
from the pore-water [CO₂] data.

Our argument is that the contributing area of the stream lateral CO₂ export is uncertain and specific
discharge within the peatland could vary greatly between areas of preferential and non-preferential flow.

Knowing that our study site is located within an area of preferential flow suggests that lateral CO₂ export
could be proportionally greater at this location compared with other areas of the peatland. While the total
stream CO₂ export is certainly small compared with the losses in porewater CO₂ store, we consider that
this pathway could still contribute to a part of the observed losses in the porewater CO₂ store. We added
more details to this paragraph (Line 360-364)

Line 346: in the catotelm

Corrected

Line 350-353: Okay, but care is needed here. Peatlands are massive C stores, but their sink strength is quite small.
The biggest concern is destabilization of the store, which occurs mainly by direct human action such as drainage,
rather than by climate change (at least currently).

We agree with the reviewer and have removed the following sentence which was misleading in this
regard: "*Although northern peatlands cover only 3% of the global land surface⁶², they play a fundamental role in*
*modulating the Earth's radiative budget^{1,2}.*" The main importance of peatlands for the current atmospheric
radiative budget is attributed to their CH₄ emissions, which could also be affected by changes in
turbulent diffusion, now mentioned in Line 379-380.

Line 355: Please consider acknowledging the reviewers.

We have gladly done so.

Line 394: consists

Corrected

Line 412: '~100 % of' (change order)

Corrected

Line 413: Provide botanical authorities for the Sphagna.

Botanical authorities now provided (Line 445-448)

Line 422: I raised this matter in my first report. If a well is screened below the lowest position of the water table
and is sealed at the surface, it will not function as a piezometer. Air trapped in the well tube will act to stop water
ingress and egress. This problem is well known among hydrogeologists. It is unclear whether this problem
produced artefacts or errors in the data. Please see my new report.

See response to general comment 2 above. Changes made on Line 457-463.

A minor point: 'bong' should be 'bung'.

Corrected

Line 439: and

Corrected

Line 455: from the

Corrected

Line 457: volume

Corrected

Line 468: Provide units.

Units provided (Line 505)

Line 480: Okay; see my comment on line 207. What is said here is right but seems to be inconsistent with what is

written on lines 207-8.

The statement has been removed (see response to general comment 3 above). This change was supported

by your comment and the reference (Rappoldt, 2003)

Line 489: Units? In a few places there appears to be confusion between units of diffusive flux (amount $m^{-2} s^{-1}$)

and the diffusivity or diffusion coefficient ($m^2 s^{-1}$). From the text it is not always clear what is intended.

We double checked all units. Diffusion coefficient are expressed in ($m^2 s^{-1}$) and kinetic energy dissipation

rate in ($m^2 s^{-3}$)

Line 497: when the

Corrected

Line 511: Italicise all variables and parameters in the equation and the accompanying text?

Done

Line 512: Explain the difference between k_C and k_{600} ? For a journal like Nature Comms it's worth considering

that non-specialists will read the paper too. Ah, I see you do this later. Perhaps refer to that later explanation

here (say something like 'see below' in brackets).

Reference to equation 5 now added (Line 550)

Line 519: But this is true of CH_4 too.

Statement changed: "largely influenced by primary production above ground".

Line 520: Better to say 'Steady ebullitive fluxes...'?

"Steady" is now added

Line 521: Supporting references?

Reference added

Line 548: 'using R (R Core Team, 2021).'

Corrected

Figure 1: Line 762: relative to

Corrected

Line 789: is in

Corrected

Line 823: compared with

Corrected

Andy Baird,
Chair of Wetland Science
University of Leeds, UK;
5th August 2021.

REVIEWERS' COMMENTS

Reviewer #3 (Remarks to the Author):

I am happy to confirm that I believe the authors have addressed my comments from 5th August satisfactorily. This is interesting work that deserves to be published. I don't quite agree with the strength of some of the arguments the authors make, but believe it is important for the paper to be in the public domain so that other scientists can debate the proposed CO₂ release mechanism.

I spotted some minor textual errors as follows:

Line 16: "propagating across" better as "propagating through"?

Line 106: "losses in CO₂ store" would be better as "losses from the CO₂ store"

Line 350: "Both of these porewater depths contain" better as "Porewater at both of these depths contains"?

Line 373: "magnitude faster" would be better as "magnitude greater"

Line 812: Typo - "vertical dotted vertical"

**Reviewer #3 (Remarks to the Author):**

I am happy to confirm that I believe the authors have addressed my comments from 5th August
satisfactorily. This is interesting work that deserves to be published. I don't quite agree with the
strength of some of the arguments the authors make, but believe it is important for the paper to be in
the public domain so that other scientists can debate the proposed CO2 release mechanism.

We thank the reviewer for his positive response and are most grateful for his effort in reviewing
our manuscript

I spotted some minor textual errors as follows:

Line 16: "propagating across" better as "propagating through"?

Corrected

Line 106: "losses in CO2 store" would be better as "losses from the CO2 store"

Corrected

Line 350: "Both of these porewater depths contain" better as "Porewater at both of these depths
contains"?

Corrected

Line 373: "magnitude faster" would be better as "magnitude greater"

Corrected

Line 812: Typo - "vertical dotted vertical"

Corrected
